# Interferon gene therapy reprograms the leukemia microenvironment inducing protective immunity to multiple tumor antigens

Giulia Escobar[1,2,3], Luigi Barbarossa[2,3], Giulia Barbiera[3], Margherita Norelli[1,4], Marco Genua[3], Anna Ranghetti[2], Tiziana Plati[3], Barbara Camisa[4], Chiara Brombin[5], Davide Cittaro [3,6], Andrea Annoni[3], Attilio Bondanza[1,4], Renato Ostuni[3], Bernhard Gentner[3,7] & Luigi Naldini[1,2,3]

Immunotherapy is emerging as a new pillar of cancer treatment with potential to cure. However, many patients still fail to respond to these therapies. Among the underlying factors, an immunosuppressive tumor microenvironment (TME) plays a major role. Here we show that monocyte-mediated gene delivery of IFNα inhibits leukemia in a mouse model. IFN gene therapy counteracts leukemia-induced expansion of immunosuppressive myeloid cells and imposes an immunostimulatory program to the TME, as shown by bulk and single-cell transcriptome analyses. This reprogramming promotes T-cell priming and effector function against multiple surrogate tumor-specific antigens, inhibiting leukemia growth in our experimental model. Durable responses are observed in a fraction of mice and are further increased combining gene therapy with checkpoint blockers. Furthermore, IFN gene therapy strongly enhances anti-tumor activity of adoptively transferred T cells engineered with tumor-specific TCR or CAR, overcoming suppressive signals in the leukemia TME. These findings warrant further investigations on the potential development of our gene therapy strategy towards clinical testing.

[1] Vita-Salute San Raffaele University, 20132 Milan, Italy. [2] Targeted Cancer Gene Therapy Unit, IRCCS San Raffaele Scientific Institute, 20132 Milan, Italy. [3] San Raffaele Telethon Institute for Gene Therapy, 20132 Milan, Italy. [4] Division of Immunology, Transplant and Infectious Diseases, IRCCS San Raffaele Scientific Institute, 20132 Milan, Italy. [5] CUSSB-University Center for Statistics in the Biomedical Sciences, Vita-Salute San Raffaele University, 20132 Milan, Italy. [6] Centre for Translational Genomics and Bioinformatics, IRCCS San Raffaele Scientific Institute, 20132 Milan, Italy. [7] Hematology and Bone Marrow Transplantation Unit, IRCCS San Raffaele Scientific Institute, 20132 Milan, Italy. These authors contributed equally: Bernhard Gentner, Luigi Naldini. Correspondence and requests for materials should be addressed to B.G. (email: gentner.bernhard@hsr.it) or to L.N. (email: naldini.luigi@hsr.it)

Increased understanding of the mechanisms co-opted by cancer cells to evade immune responses has led to the development of novel therapeutics targeting immune checkpoints[1]. Clinical testing of these drugs has led to unprecedented rates of durable responses in several types of tumors[2,3]. However, despite these advances, a large fraction of patients do not respond to these therapies, due to the failure to generate tumor-specific T cells and the existence of an immunosuppressive TME, which imparts resistance to blockade of the classical checkpoints, CTLA4 or PD1/PDL1[4]. Current efforts are thus aiming at identifying new immune checkpoint targets and combination therapies, which might extend the benefits of immunotherapy to a larger number of patients. Another immunotherapeutic approach showing promising results in the clinics is the adoptive transfer of genetically engineered T cells expressing a transgenic T cell (TCR) or chimeric antigen receptor (CAR) directed against a tumor-specific antigen (TSA)[5,6]. This strategy is particularly suitable for malignancies with low mutation burden that fail to induce endogenous T cell responses against TSAs. CAR T cells recognizing the CD19 antigen have demonstrated remarkable efficacy in relapsed and refractory B cell malignancies. However, these studies also suggested that the therapeutic effect was less evident in nodal disease with respect to bone marrow (BM) disease or leukemia, suggesting that an immunosuppressive TME represents a major impediment towards successful immunotherapy, especially against solid tumor masses. Moreover, in fast-growing tumors such as B cell acute lymphoblastic leukemia (B-ALL), antigen loss occurs in 20% of patients treated with CD19 CAR T cells, highlighting a limitation of immunotherapy directed against a single antigen[5,7].

Recently, there has been renewed interest in the use of type-I interferons (IFNs) as anti-cancer agents[8]. In addition to the cytostatic and anti-angiogenic effects on tumor cells and blood vessels, type-I IFNs increase the maturation and cross-priming capacity of dendritic cells (DCs), the proliferation and cytotoxicity of T cells, the killing capacity of NK cells, and immunoglobulin class switching of B cells[9,10]. We previously reported proof-of-principle that a cell and gene therapy strategy selectively expressing an IFNα transgene in the TIE2 + tumor infiltrating monocyte/macrophage progeny of transplanted, genetically engineered hematopoietic stem cells (HSC) can induce relevant anti-tumor responses. This monocyte-mediated IFN gene therapy showed no systemic toxicity in the mice and inhibited the growth of spontaneous mammary tumors as well as lung and liver metastases of breast and colorectal cancer cells, respectively[11–13]. Even though we provided some evidence for immune-mediated effects in these studies, whether IFN gene therapy can engage the tumor-immunity equilibrium and support deployment of adaptive immunity remains to be determined. Here we exploited a novel, immune-competent mouse model mimicking human B-ALL[14] and show that monocyte-mediated IFNα delivery can reprogram the TME towards inducing effective anti-tumor immune responses and synergizes with checkpoint blockade and adoptive T-cell immunotherapies in the treatment of a disseminated hematologic malignancy.

## Results

**IFN gene therapy boosts T cell immunity in a B-ALL model.** We transplanted C57Bl/6 mice with HSC transduced with either *mTie2*-IFN-mirT LV (IFN mice) or *mTie2*-GFP-mirT LV or Mock-transduced (both used as control mice), to target IFN/GFP expression to the differentiated TIE2 + monocyte progeny, which is highly enriched in tumors[15,16]. As shown previously, reconstitution with *mTie2*-IFN-mirT LV transduced cells results in a functional multi-lineage graft, with no overt side effects[11–13]. We

then challenged reconstituted mice with our previously described B-ALL model (Fig. 1a) and found inhibition of leukemia growth in IFN vs. control mice (Fig. 1b, c, *$p < 0.05$, nonparametric rank-based method for longitudinal data in factorial experiments). Administration of anti-CTLA4 blocking antibody (αCTLA4) had no effects in control mice but further and significantly improved ALL inhibition in IFN mice, suggesting an immune contribution to the observed response in IFN mice (IFN vs IFN + αCTLA4 **$p < 0.01$, IFN vs CTRL + αCTLA4 *$p < 0.05$, IFN + αCTLA4 vs CTRL + αCTLA4 ****$p < 0.0001$, CTRL vs CTRL + αCTLA4 n.s., nonparametric rank-based method for longitudinal data in factorial experiments). The combination of IFN gene therapy and αCTLA4 significantly improved the survival of the mice (Supplementary Fig. 1a, **$p < 0.01$ Mantel–Haenszel test). To investigate the mechanisms by which IFN gene therapy contributes to the induction of anti-tumor immunity, we engineered ALL cells with a LV allowing coordinate expression of the Ovalbumin (OVA) model antigen and the truncated form of the nerve growth factor receptor (NGFR) cell surface marker from a bidirectional promoter (OVA-ALL, Supplementary Fig. 1b and c). When injected into immunocompetent C57Bl/6 mice, OVA-ALL showed slower growth kinetics and delayed onset in a fraction of the mice as compared to parental ALL (Supplementary Fig. 1d). At necropsy, all mice showed massive BM infiltration by ALL cells, with outgrowth of NGFR-negative blasts in the mice showing delayed disease (Supplementary Fig. 1f). When OVA-ALL cells were injected in immunodeficient NOD-SCID-IL2Rg−/− (NSG) mice, we observed comparable tumor growth as the parental cells and no loss of NGFR expression (Supplementary Fig. 1e, g). Altogether, these results indicate increased immunogenicity of the OVA-ALL variant leading to immune editing and selection of rare un-transduced or silenced (OVA/NGFR negative) ALL clones likely present in the infusion product. No mice, however, survived either challenge. We thus tested the ability of tumor-targeted IFN gene therapy to boost the anti-tumor immune response against OVA-ALL. Whereas ALL rapidly expanded in control mice (Fig. 1d), there were delayed appearance and accumulation of blasts in the blood, and reduced infiltration in the BM and spleen of IFN mice (blood: ****$p < 0.0001$, nonparametric rank-based method for longitudinal data in factorial experiments; spleen and BM: *$p < 0.05$ and ** $p < 0.01$ respectively, Mann–Whitney). Of note, a fraction of IFN mice showed absence of leukemia in all organs analyzed. In vitro stimulated, purified splenic CD8+ T cells from IFN mice showed induction of a specific response against OVA by γ-IFN-ELISPOT, and the mice with higher number of responder cells showed the lowest tumor burden (Supplementary Fig. 2a). OVA-specific CD8+ T cells were detected by OVA$_{257–264}$-H-2K$^b$-pentamer staining in both IFN and control mice challenged with ALL, with higher percentages and numbers in the blood and BM of the former group (Fig. 2a, b and Supplementary Fig. 2b. Blood: **$p < 0.01$; BM: ***$p < 0.001$, Mann–Whitney). In contrast with the T cells of IFN mice, CD8+ T cells of control mice did not release IFN-γ upon ex-vivo re-stimulation with OVA, suggesting T cell dysfunction, consistently with the lack of tumor inhibition in control vs. IFN mice (Fig. 2c and Supplementary Fig. 2c-e). Strikingly, depletion of CD8 + T cells in IFN mice abrogated the anti-tumor response (Fig. 2d and Supplementary Fig. 2f. IFN vs IFN + αCD8 **$p < 0.01$, IFN vs CTRL ****$p < 0.0001$, IFN + αCD8 vs CTRL n.s., nonparametric rank-based method for longitudinal data in factorial experiments). Overall, these results indicate induction of CTLs able to mount an effective response against a TAA in IFN mice.

We also noted an initial lower tumor burden in CD8-depleted IFN mice as compared to control (Supplementary Fig. 2g), suggesting that additional mechanisms beside CD8-mediated control may contribute, at least initially and temporarily, to

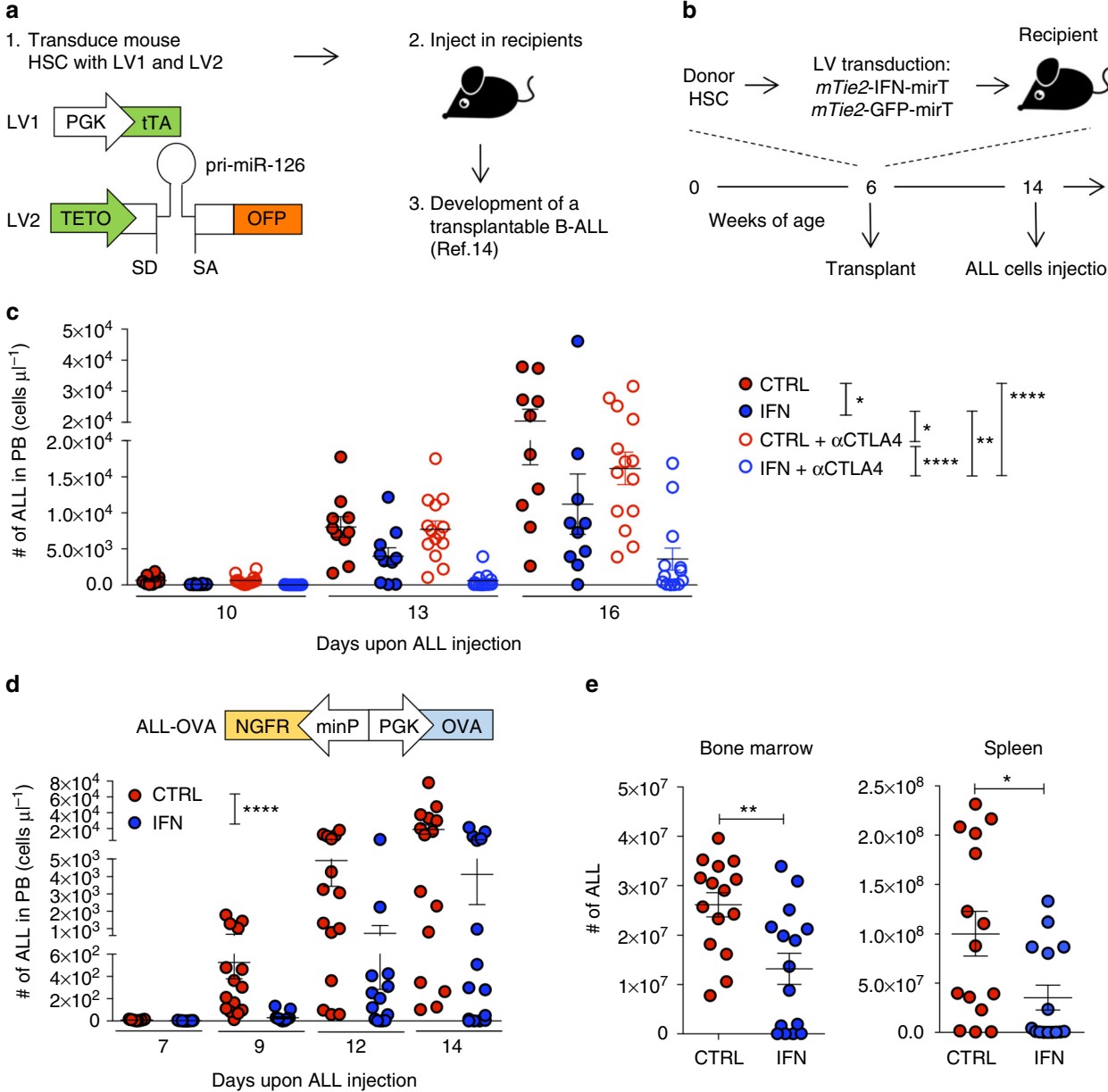

**Fig. 1** IFN gene therapy inhibits ALL growth. **a** Schematic representation of the lentiviral vectors (LV) used to engineer HSC and generate spontaneous B-ALL. **b** Experimental design. **c** Absolute numbers (mean ± SEM) of parental ALL in the peripheral blood (PB) over time of CTRL ($n = 10$, treated with isotype control antibody), IFN ($n = 10$, treated with isotype control antibody), CTRL + αCTLA4 ($n = 14$) and IFN + αCTLA4 ($n = 13$) mice. Each dot represents a mouse. $*p < 0.05$, $**p < 0.01$, $****p < 0.0001$, nonparametric rank-based method for longitudinal data in factorial experiments. **d, e** Absolute numbers (mean ± SEM) of OVA-ALL (engineered with the NGFR/OVA bidirectional LV shown on the left) in the PB (**d**) over time and in BM and spleen (**e**, 14 days after tumor injection) of IFN ($n = 15$) and CTRL ($n = 15$) mice (one out of 9 experiments is shown). Each dot represents a mouse. $****p < 0.0001$, nonparametric rank-based method for longitudinal data in factorial experiments; $*p < 0.05$, $**p < 0.01$, Mann–Whitney

tumor inhibition. At this early time-point, we found no difference in the apoptotic cell fraction and cell cycle distribution of BM and spleen ALL cells between IFN and control mice (Supplementary Fig. 3a–d), while the proliferation rate, as measured by EdU incorporation, was lower in the BM, but not the spleen, of IFN mice (Supplementary Fig. 3e, f, $*p < 0.05$, Mann–Whitney). Thus, a delay in proliferation, likely triggered directly by IFN, may favor the buildup of tumor specific CTL at effective effector to target ratio to suppress tumor cell growth.

**Monocytes-mediated IFN delivery reprograms the TME**. To study whether the induction of effective immune responses by IFN gene therapy was associated with changes in the leukemia TME,

we performed immunophenotypic analyses on subpopulations of innate and adaptive immune cells from the blood, spleen and BM. Leukemia induced a dramatic increase in the percentage of non-classical (Ly6C-Ly6G-) monocytes as well as a reduction of classical (Ly6C + Ly6Gint) monocytes in the blood (Fig. 3a and Supplementary Fig. 4a). These changes were nearly abrogated in IFN mice (IFN + ALL vs CTRL + ALL: non classical monocytes $***p < 0.001$, classical monocytes $****p < 0.0001$, Kruskal–Wallis and Dunn correction). Leukemia growth in the spleen was accompanied by expansion of non-classical monocytes, which likely comprise immature myeloid derived suppressor cells (iMDSC) (Fig. 3b and Supplementary Fig. 4b. CTRL vs CTRL + ALL $*p < 0.05$, Kruskal–Wallis and Dunn correction), and

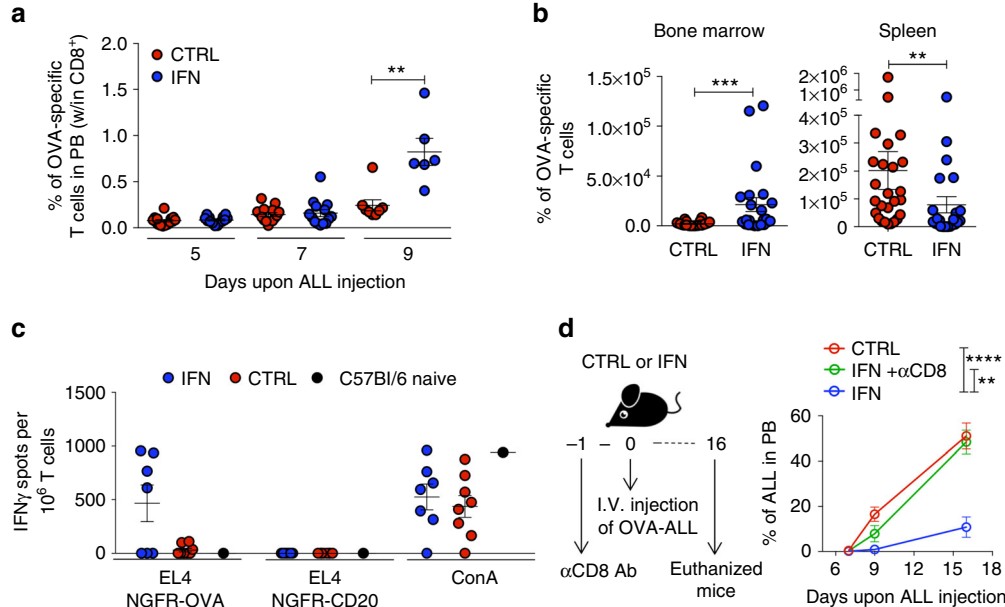

**Fig. 2** IFN gene therapy activates adaptive immunity. **a** Percentage (mean ± SEM) of OVA-specific T cells in the PB of IFN (*n* = 14–6) and CTRL (*n* = 15–8) mice. \*\**p* < 0.01, Mann Whitney performed at 9 days upon OVA-ALL injection. Each dot represents a mouse (one out of five experiments is shown). **b** Absolute numbers (mean±SEM) of OVA-specific T cells in the BM and spleen, 9 days upon tumor injection, in IFN (*n* = 23) and CTRL (*n* = 26) mice from 3 independent experiments. \*\**p* < 0.01, \*\*\**p* < 0.001, Mann–Whitney. Each dot represents a mouse. **c** Splenic CD8+ T cells from IFN (*n* = 7) and CTRL (*n* = 8) mice 9 days upon tumor injection or from non tumor-bearing mice tested by IFNγ-ELISPOT against the EL4 target cell line transduced with NGFR-OVA or NGFR-hCD20 bidirectional LV. Concanavalin A (ConA) stimulation is used as positive control. Each dot represents a mouse. **d** Experimental design and percentage (mean±SEM) of OVA-ALL in the PB of IFN (*n* = 9), CD8-depleted IFN (*n* = 7) and CTRL (*n* = 9) mice. \*\**p* < 0.01, \*\*\*\**p* < 0.0001, nonparametric rank-based method for longitudinal data in factorial experiments

increased percentage of MHC-II-negative macrophages (Fig. 3c, CTRL vs CTRL + ALL \*\**p* < 0.01, Kruskal–Wallis and Dunn correction). Such changes were not seen in IFN mice. In the BM, leukemia induced a substantial depletion of myeloid cells (IFN + ALL vs CTRL + ALL: macrophages \*\**p* < 0.01, neutrophils \**p* < 0.05, non classical monocytes \**p* < 0.05, Kruskal–Wallis and Dunn correction), an effect not seen in IFN mice, which instead showed an increase in the number of DCs and in the fraction of DC presenting the immune-dominant OVA$_{257-264}$ peptide on MHC-I (Fig. 3d, e and Supplementary Fig. 4c. IFN + ALL vs CTRL + ALL: (e) \*\**p* < 0.01, Kruskal–Wallis and Dunn correction; (e) \**p* < 0.05, Mann–Whitney).

RNA-sequencing (RNA-Seq) analyses revealed leukemia-induced transcriptional changes in macrophages, which were substantially counteracted by IFN gene therapy (Supplementary Fig. 5a, b and Supplementary Data 1 and 2). Spleen macrophages from ALL vs. control mice up-regulated genes encoding for the immunosuppressive cytokine IL-10, the inhibitory immune checkpoint PD-1, as well as genes linked to cell division and response to immune stimuli gene ontology (GO) terms. Down-regulated genes were enriched in GO terms related to fatty acid metabolism, leukocyte activation and antigen presentation (Fig. 4a, c and Supplementary Data 3). IFN gene therapy in ALL mice elicited an immunostimulatory program characterized by up-regulation of IFN-Stimulated Genes (ISGs) enriched in defense response, leukocyte migration and response to interferon GO terms, and abrogated leukemia-induced up-regulation of *Il10* and down-regulation of MHC II genes (Fig. 4b–d and Supplementary Data 3). IFN gene therapy in ALL mice induced ISGs at levels higher than those triggered in controls, (Fig. 4d and Supplementary Fig. 5a), and the transcriptomes of macrophages from control and IFN tumor-free mice showed high correlation, while they were clearly distinct from the ALL and IFN+ALL groups (Supplementary Fig. 5b). These data confirm and extend

previous reports that our monocyte-mediated gene therapy preferentially targets IFN to the TME[11–13].

To dissect the impact of the leukemia and IFN gene therapy on the TME in a more unbiased manner, we performed single-cell (sc)RNA-Seq on CD11b + cells isolated from the spleen of control and tumor-bearing mice, treated or not with IFN gene therapy. Using a droplet-based approach (PMID 28091601), we generated scRNA-Seq data from 10,821 cells, detecting a mean of 1,338 genes/cell (Supplementary Data 1). Graph-based clustering and gene signature analyses using published datasets (PMID 25480296, 28636953, ImmGen) identified 11 clusters corresponding to monocytes (cl. 1–3), neutrophils (cl. 4–6), dendritic cells (cl. 7), macrophages (cl. 8), natural killer and T cells (cl. 9), mast cells (cl. 10), and B cells (cl. 11). Heterogeneity was observed within the monocyte and neutrophil populations, encompassing non-classical (cl. 1) and classical (cl. 2) monocytes, a cluster co-expressing monocyte and neutrophil genes (cl. 3) (PMID 29166589) and neutrophil maturation intermediates (Fig. 5a, Supplementary Fig. 6a, b and Supplementary Data 4). Leukemia had a major impact on the transcriptional landscape of non-classical monocytes (Fig. 5b and Supplementary Data 5), which were expanded in the spleen of tumor-bearing mice (see Fig. 3b). The other myeloid cell populations, including macrophages and DCs, showed comparatively less leukemia-induced alterations (Supplementary Fig. 7). Tumor-associated non-classical monocytes up-regulated genes enriched in GO terms such as complement activation and negative regulation of inflammation, while they down-regulated genes linked to antigen processing and presentation (Fig. 5c and Supplementary Data 6). IFN gene therapy imposed an ISG-driven immunostimulatory program to non-classical monocytes from ALL mice, as evidenced by up-regulation of genes enriched in GO terms related to defense and innate immune response, as well as MHC II genes (Fig. 5c, d and Supplementary Data 6).

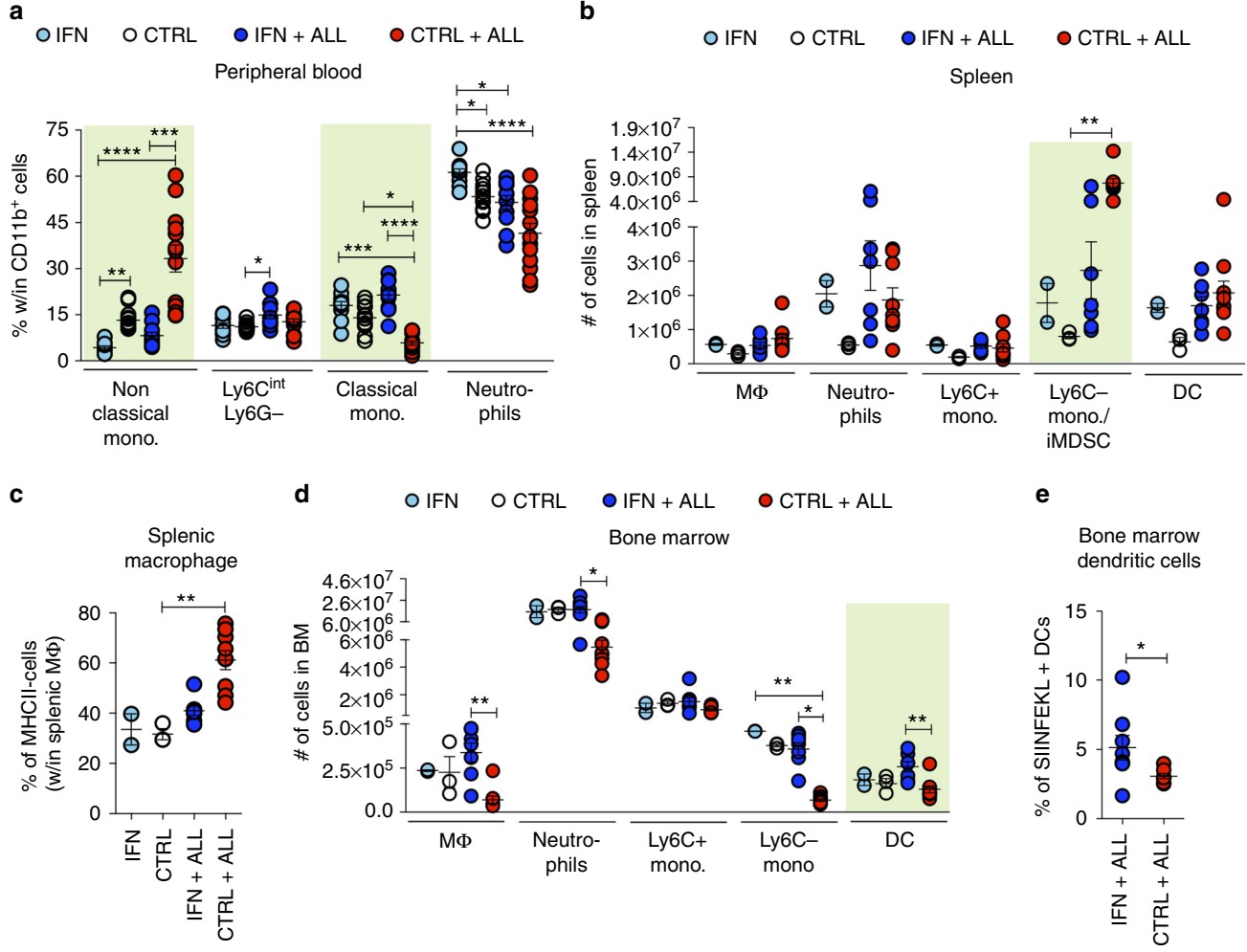

**Fig. 3** IFN gene therapy imposes an immune-stimulatory program in the TME. **a** Percentage of the indicated populations within total myeloid cells in the PB of IFN ($n = 11$) and CTRL ($n = 13$) mice before and after (12 days) OVA-ALL injection. $*p < 0.05$, $**p < 0.01$, $***p < 0.001$, $****p < 0.0001$, Kruskal–Wallis, with adjusted $p$-value by Dunn's test. Statistical analysis is performed within each individual myeloid cell population. **b** Absolute numbers (mean ± SEM) of the indicated populations in the spleen of tumor-free and OVA-ALL-injected IFN and CTRL mice (CTRL: $n = 3$, IFN $n = 2$, CTRL + ALL: $n = 9$, IFN + ALL: $n = 7$), 10 days upon tumor injection. Each dot represents a mouse. $**p < 0.01$ Kruskal-Wallis with adjusted $p$-value by Dunn's test. Statistics is performed within each individual population. **c** Percentage of MHC-II- macrophages present in tumor-free or tumor-bearing IFN and CTRL mice from **b**. $**p < 0.01$, Kruskal–Wallis with adjusted $p$-value by Dunn's test. **d** Absolute numbers (mean±SEM) of the indicated populations in the BM of tumor-free and OVA-ALL-injected IFN and CTRL mice from **b**. Each dot represents a mouse. $*p < 0.05$, $**p < 0.01$, Kruskal–Wallis with adjusted $p$-value by Dunn's test. Statistics is performed within each individual population. **e** Percentage of DCs presenting the immune-dominant OVA (SIINFEKL) peptide on MHC-I molecules in the BM of mice from CTRL+ALL ($n = 7$) and IFN + ALL ($n = 8$). $*p < 0.05$ Mann–Whitney test

Transcriptional reprogramming of the TME by IFN gene therapy was less effective in non-classical monocytes from mice that did not respond to IFN gene therapy (Supplementary Fig. 8a), as revealed by graph-based clustering and differential gene expression (Fig. 5b, d and Supplementary Data 5). Subclustering of scRNA-Seq data from non-classical monocytes identified four major subclusters (1A to 1D, Supplementary Fig. 8b). Subcluster 1A was comprised of cells from disease-free mice from both control and IFN mice, whereas the other three subclusters largely overlapped with cells from IFN+ALL responder (1B), IFN+ALL non responder (1C), and ALL (1D). Minimum-spanning tree (MST) analyses revealed a trajectory from 1A to 1D, confirming partial vs. effective reprogramming in cells from non responder vs. responder IFN mice (Fig. 5b left panel).

Overall, these data indicate that IFN gene therapy imposes an immunostimulatory program to the myeloid cell infiltrate, conceivably priming towards activation of Th1 responses. Indeed, purified splenic CD4$^+$ T cells from IFN+ALL mice showed significant up-regulation of *Tbx21*, encoding the Th1 transcription factor TBET, and *Il17*, a prototypical Th17 gene, while there were no changes in prototypical Th2 and Treg genes (Supplementary 8c. *Tbx1* and *Il17*, $*p < 0.05$, Mann–Whitney).

**IFN induces protective immunity by targeting multiple surrogate TSAs.** We then investigated whether tumor-targeted IFN gene therapy could promote durable responses in the mice. Strikingly, an average of 24% (5 different experiments, $n = 11, 14, 8, 16, 12$) of IFN mice survived long-term and were cured of the disease, whereas only 3% of control mice ($n = 13, 14, 8, 16, 13$) survived to it (a representative experiment is shown in Fig. 6a and Supplementary Fig. 9a. $*p < 0.05$ Mantel–Haenszel test). Of note, by stratifying the mice based on survival time, we found that mice euthanized early after tumor injection showed low numbers of circulating OVA-specific T cells and no appearance of NGFR-

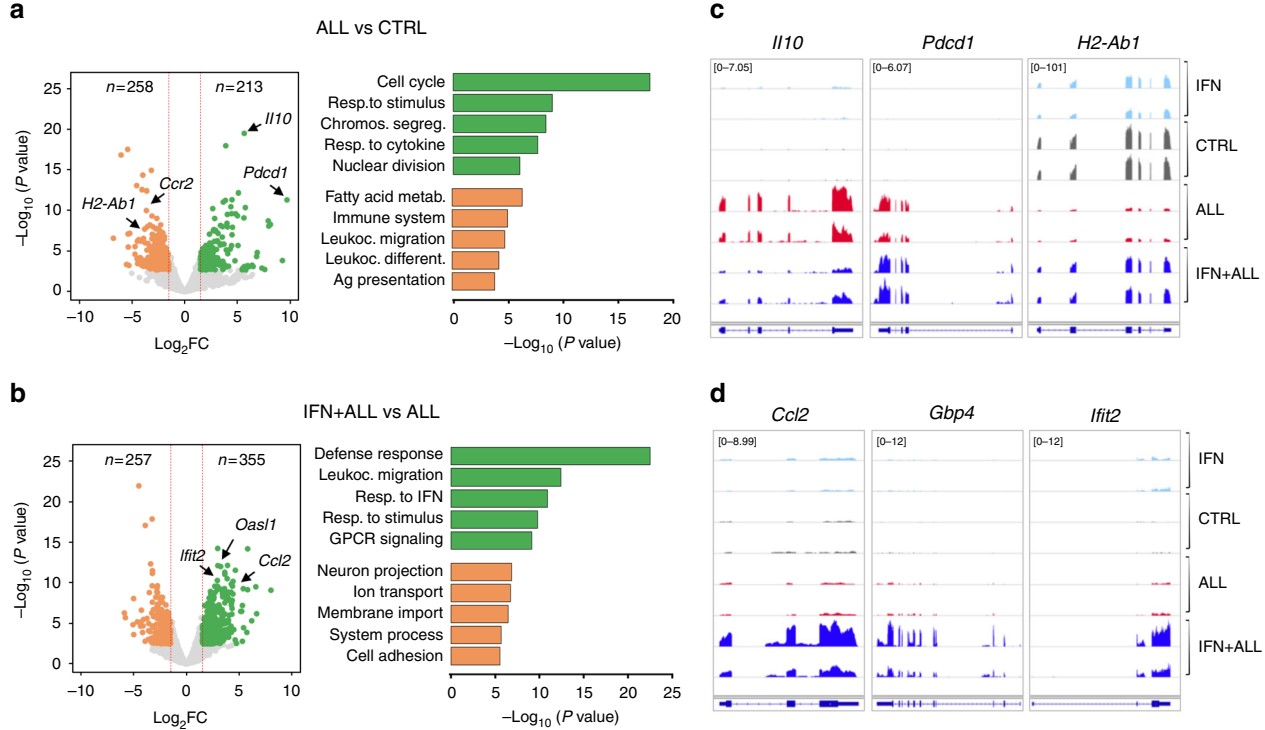

**Fig. 4** Transcriptional reprogramming of tumor-associated macrophages by IFN gene therapy. **a**, **b** Volcano plots (left panels) showing differentially expressed genes (abs. log$_2$FC > 1.5, FDR < 0.05) up-regulated (green) or down-regulated (orange) in splenic macrophages from tumor-bearing (ALL) vs. control (CTRL) mice (**a**) or ALL mice treated (IFN+ALL) or not (ALL) with IFN gene therapy (**b**). Bar plots (right panel) show GO terms enriched in up-regulated (green) or down-regulated (orange) genes from the indicated comparisons. **c**, **d** Selected RNA-Seq snapshots of deregulated genes in macrophages from ALL mice (**c**) or ISGs induced in IFN+ALL mice (**d**)

negative ALL (Fig. 6b). Conversely, mice euthanized at later time points showed increasing numbers of circulating OVA-specific T cells and the appearance of NGFR-negative ALL clones. These results suggest that mice succumbing to the disease either failed to mount an immune response and died very early, or mounted anti-OVA responses but ultimately died due to the failure of these cells to protect the mice, either because of exhaustion or the emergence of immune-selected OVA-negative ALL clones. Long-term survivors, exclusively present in the IFN group, showed circulating OVA-specific T cells (Fig. 6b, right most panel) and, when re-challenged with OVA-ALL, remained disease-free (Fig. 6c). Tumor clearance was associated with further expansion of circulating OVA-specific T cells, suggesting that development of memory T cell responses protected the mice from subsequent tumor challenge (Supplementary Fig. 9b). Strikingly, surviving IFN mice efficiently cleared both OVA-expressing and parental OVA-negative ALL cells, when re-challenged with a 1 to 1 ratio of these cells, or parental cells alone, suggesting spreading of the response towards additional TSAs (Fig. 6d, e and Supplementary Fig. 9c). Beside the dominant OVA antigen, our ALL model also expresses OFP and the prokaryotic trans-activator tTA (see Fig. 1a), which can serve as surrogate TSA. We thus performed a new experiment, harvested peripheral blood mononuclear cells (PBMCs) of control and IFN mice early after leukemia injection, stimulated them with target cells transduced with LV expressing tTA, OFP, or OVA, and measured γ-IFN production by ELISPOT assay (Fig. 6f and Supplementary Figure 16). Intriguingly, the two IFN mice eventually surviving the tumor challenge showed early reactivity against both OVA and tTA, suggesting that immune response to multiple surrogate TSAs might be required to achieve durable protection.

**IFN delivery improves efficacy of CTLA-4 blockade therapy.** Because we showed synergy between IFN gene therapy and CTLA-4 blockade against a challenge with parental ALL cells (see Fig. 1c), we tested the combination therapy also against OVA-ALL cells. Whereas both treatments alone significantly increased mouse survival vs. control, the combination therapy was more effective leading to a sizable fraction of cured mice (Fig. 7a and Supplementary Fig. 9d. IFN vs CTRL *p < 0.05, CTRL + αCTLA4 vs CTRL *p < 0.05, IFN + αCTLA4 vs CTRL ***p < 0.001, Mantel–Haenszel test and Bonferroni correction), as confirmed in a second experiment (see Fig. 7d below and Supplementary Fig. 9e. IFN + αCTLA4 vs CTRL + αCTLA4 *p < 0.05, Mantel–Haenszel test). IFN gene therapy or αCTLA-4 treatment increased the percentage of circulating OVA-specific T cells, which further increased in the combination group (Fig. 7b. IFN vs CTRL *p < 0.05, CTRL + αCTLA4 vs CTRL **p < 0.01, IFN + αCTLA4 vs CTRL ****p < 0.0001, nonparametric rank-based method for longitudinal data in factorial experiments). Accordingly, immune selection of NGFR-negative ALL was more evident in the IFN + αCTLA4 and αCTLA4 groups (Supplementary Fig. 10a). We then assessed the immune reactivity of PBMCs from surviving mice and found response against one or more TSA in addition to OVA in all of them (Supplementary Fig. 11a and Supplementary Figure 16). We re-challenged these mice with a 1:1 ratio of OVA-positive and negative ALL and found that most of them survived, whereas those failing to survive succumbed to the selection of ALL clones lacking more than one surrogate TSA (Supplementary Figure 16). As a further indication of an adaptive immune response underlying the survival of the mice, we compared the TCR-beta complementary determining region (CDR) repertoire of PBMCs before and after the leukemia

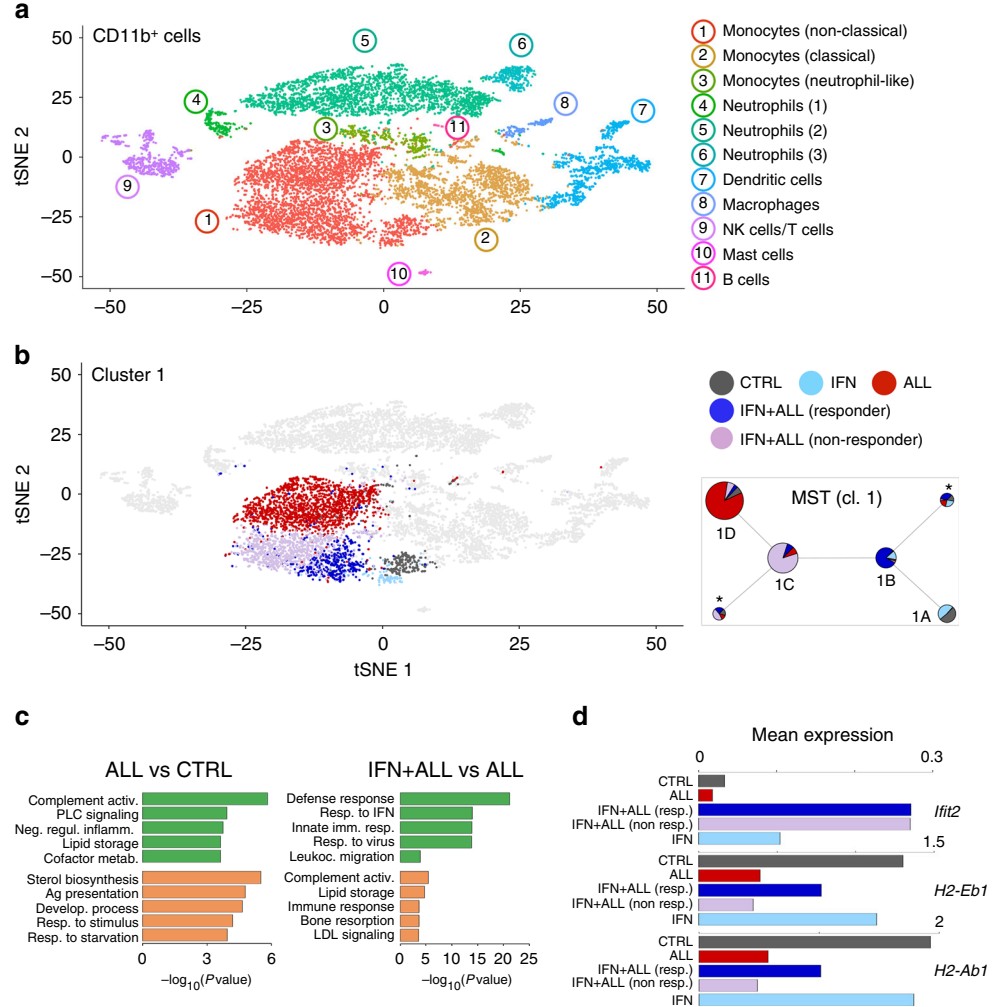

**Fig. 5** Single cell transcriptional reprogramming of tumor-associated myeloid cells by IFN gene therapy. **a** tSNE plots incorporating scRNA-Seq data from 10,821 CD11b$^+$ cells sorted from the spleen of CTRL or ALL mice, treated or not with IFN gene therapy. Transcriptionally defined clusters and associated cell types are indicated by numbers and colors in the legend. **b** Right plot: tSNE plots incorporating single-cell RNA-Seq data of cells from cluster 1 (non-classical monocytes), colored based on the experimental condition. Left plot: Minimum spanning tree (MST) analysis of scRNA-Seq data from cluster 1 (non-classical monocytes), after sub-clustering. Each pie represents a sub-cluster and shows its size (number of cells) and composition (experimental condition). Pies marked by asterisks represent minor sub-clusters with dispersed composition. **c** GO terms enriched in the top 100 genes (ranked by log$_2$FC) up-regulated (green) or down-regulated (orange) in scRNA-Seq data from cluster 1 (non-classical monocytes) in the indicated comparisons. **d** Mean expression (log transformed TPM values, normalized for number of cells) in non-classical monocytes of selected genes for the indicated conditions

challenges. Both the productive clonality (a measure of diversity ranging from 0 = polyclonal to 1 = monoclonal within each mouse) and the similarity of the repertoire among different mice increased upon leukemia re-challenge, indicating the expansion of tumor-reactive T cell clones against a common set of TSAs in surviving mice (Fig. 7c and Supplementary Fig. 11b). To further assess whether generation of responses towards multiple surrogate TSAs is a predictor of long-term survival, we investigated T cell reactivity in PBMCs harvested from all mice early after OVA-ALL challenge (Fig. 7d). Mice showing an anti-tumor repertoire encompassing multiple surrogate TSAs had a much higher like-lihood of long-term survival than mice responding to a single or no surrogate TSA, which all died of the disease (Fig. 7e and Supplementary Fig. 11c).

**OT-I T cells expand and contain leukemia only in IFN mice**. To investigate the effect of IFN gene therapy on tumor-specific T cell recruitment and activation, we adoptively transferred naive transgenic OVA-specific T cells (OT-I) in IFN and control mice

(Fig. 7f). We adjusted the time of infusion between the two groups to infuse OT-I cells at comparable leukemia burden (Fig. 7g and Supplementary Fig. 12a) and analyzed mice 3 days after. We observed substantially higher numbers of OT-I cells in the spleen and BM of leukemia-injected IFN than control mice (Fig. 7h, spleen and BM: *$p < 0.05$ and **$p < 0.01$ respectively, Mann-Whitney). In both groups of mice, OT-I cells up-regulated the activating receptor LAG3, and acquired central or effector memory phenotype, whereas OT-I cells maintained the naive phenotype of the harvest in tumor-free control mice (Supplementary Fig. 12b, c). Leukemia burden was significantly reduced in IFN vs. control mice already at early times after adoptive T cell transfer (Supplementary Fig. 12d. BM and spleen: **$p < 0.01$ and ***$p < 0.001$ respectively, Mann–Whitney). Whereas nearly all leukemic cells of control mice expressed the NGFR marker, there was an increasing fraction of NGFR-negative cells in IFN mice, indicating selective pressure against OVA-ALL (Supplementary Fig. 12e). We then explored the synergy of IFN gene therapy and OT-I T cells in promoting survival (Fig. 7i, right panel). Whereas adoptive transfer of OT-I T cells resulted in 20% mice survival vs

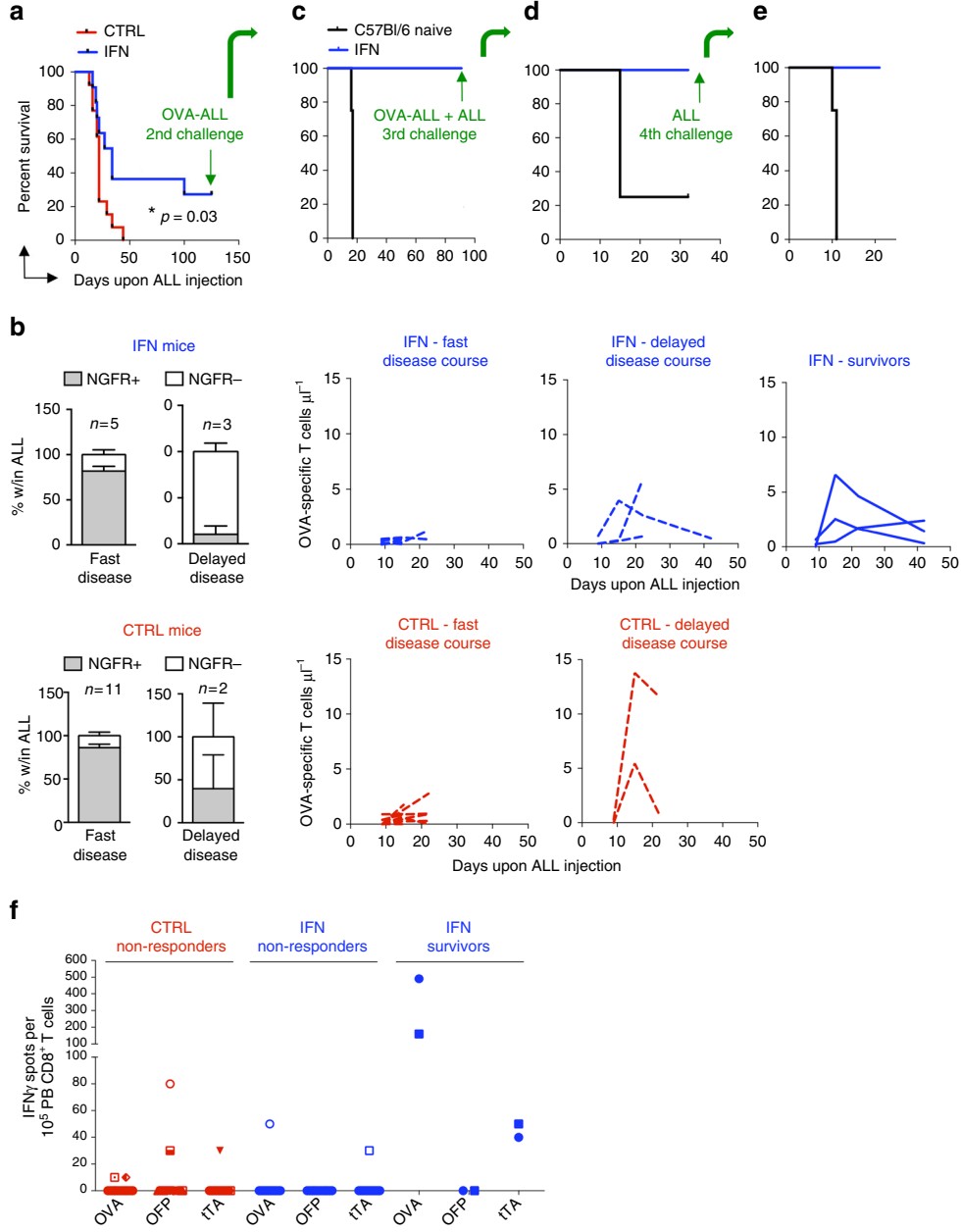

**Fig. 6** IFN gene therapy promotes immune reactivity towards multiple surrogate TSAs and survival in a fraction of mice. **a** Survival curve of OVA-ALL-injected IFN ($n = 11$) and CTRL ($n = 13$) mice after the first tumor challenge with OVA-ALL cells. *$p < 0.05$, Mantel–Haenszel test. **b** NGFR expression on BM-infiltrating ALL (or PB ALL for those mice for which BM analysis is not available) and absolute numbers (mean ± SEM) of OVA-specific T cells (each line represents a mouse) in the PB of IFN ($n = 8$) and CTRL ($n = 13$) mice showing fast or delayed disease course (fast disease course: CTRL: 13–29 days (range), 20.4 ± 1.3 (mean ± SEM), $n = 11$; IFN:16–27 days, 20.8 ± 1.8, $n = 5$; delayed disease course: CTRL: 34–44 days, 39 ± 5, $n = 2$; IFN: 34–100 days, 56 ± 22, $n = 3$) and of long term surviving IFN ($n = 3$) mice. **c**, **d**, **e** Survival curves of long-term surviving IFN mice from (**a**) subjected to subsequent tumor challenges with the indicated ALL cells (**c**: $n = 3$ long-term surviving IFN mice from **a** vs. 4 naive mice; **d**: same 3 surviving mice from **a**, **c** vs. 4 naive mice; **e**: 2 of the 3 surviving mice from **a**, **c**, **d** vs 4 naive mice. **f** PBMC from OVA-ALL injected IFN (non-responders, $n = 14$; long-term survivors, $n = 2$) and CTRL (non-responders, $n = 16$) mice tested by IFNγ-ELISPOT (13 days upon tumor injection) against the EL4 target cell line transduced with the NGFR-OVA bidirectional LV or PGK-OFP or PGK-tTA LV. Each dot represents a mouse tested against all three surrogate tumor-specific antigens

none in the control group, the combined treatment significantly improved survival to 67% (Fig. 7i. IFN + OT-I vs CTRL ****$p < 0.0001$, IFN + OT-I vs CTRL + OT-I *$p < 0.05$, Mantel–Haenszel test and Bonferroni correction). Longitudinal analysis of IFN + OT-I mice revealed OT-I T cells expansion, peaking at 6 days upon infusion, accompanied by transient LAG3 up-regulation and OVA-ALL clearance (Fig. 7j, k and Supplementary Fig. 13a). Of note, those IFN + OT-I mice that still succumbed to the disease (shown in black in Fig. 7j, k) showed the outgrowth of

NGFR-negative ALL cells in the blood and BM, thus confirming that OVA-expressing leukemic cells had been eradicated by the infused OT-I cells (Supplementary Fig. 13a, c). Conversely, OT-I T cells in most control mice failed to expand and showed constitutive high levels of LAG3 expression, a sign of T cell exhaustion (Fig. 7j, k). Accordingly, infused OT-I cells failed to eradicate OVA-expressing ALL cells in these mice, as shown by an initial but transient decrease in circulating NGFR-expressing leukemic cells followed by their outgrowth in the

blood and BM (Supplementary Fig. 13b, d). At necropsy, whereas a good fraction of OT-I T cells in the BM, spleen and lymph nodes of IFN + OT-I mice remained as memory pool and had down-regulated the PD1 inhibitory marker, most OT-I T cells in CTRL + OT-I mice had features of exhausted effectors, as they all expressed PD1 (Supplementary Fig. 13e, f). Overall, these data show that, while OT-I T cells underwent robust activation and expansion in IFN mice, leading to tumor clearance, these cells were hypo-functional in control mice, failed to expand and showed phenotypic evidence of exhaustion.

**IFN gene therapy boosts efficacy of CART19 cells**. To further explore the synergy between IFN gene therapy and adoptive T-cell transfer in a clinically relevant model, we generated T cells expressing a previously described second generation (2 G) CAR targeting the mouse CD19 (CART19 cells) and incorporating the CD28 endocostimulatory domain[17], and treated mice injected with the parental (OVA-negative) ALL. For CAR gene transfer we exploited LV that coordinately express the CAR and the NGFR marker from a bidirectional promoter[18] (Supplementary Fig. 14a, b). To allow CART19 cells engraftment[19], mice were conditioned with cyclophosphamide prior to infusion (Fig. 8a). Whereas

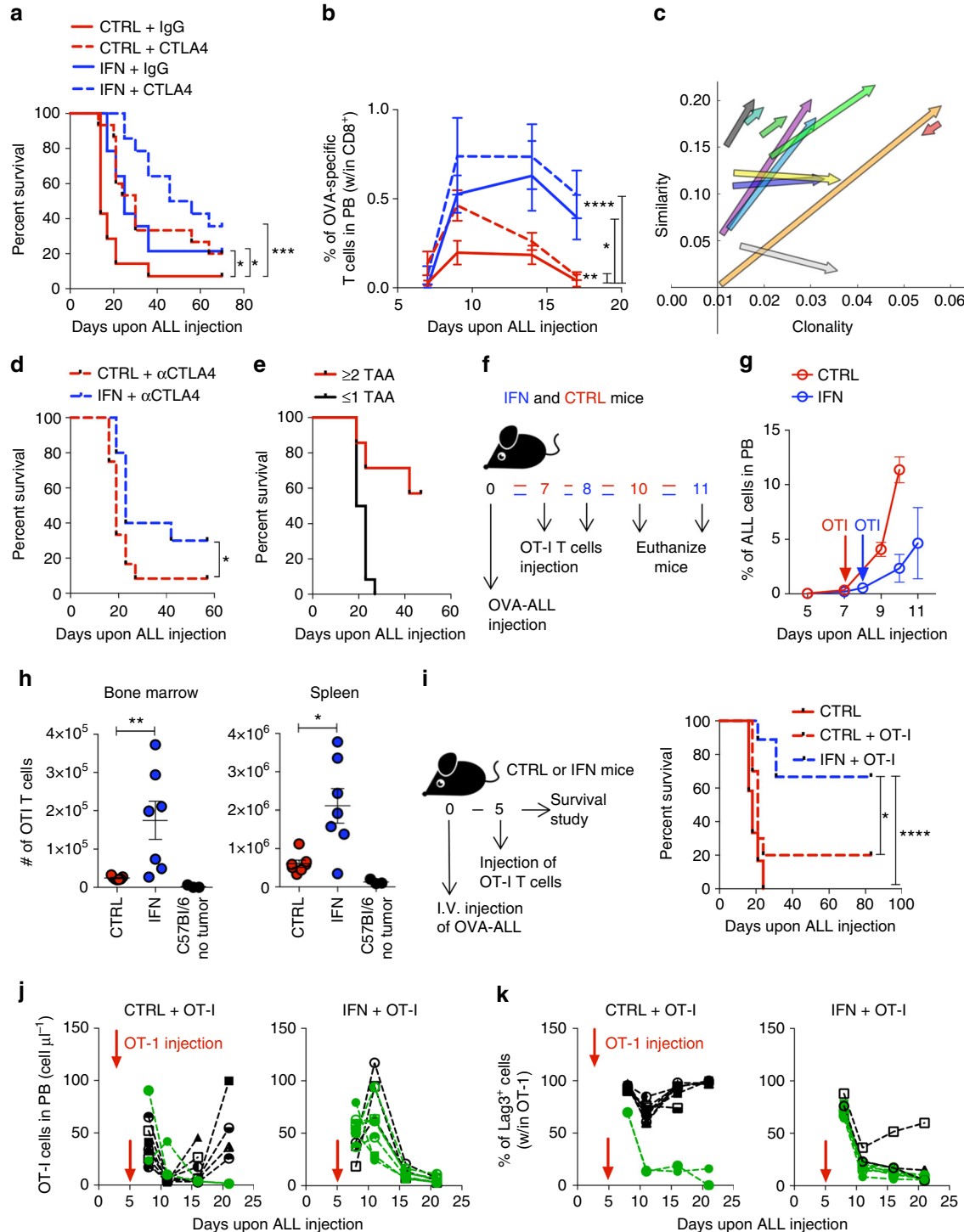

CART19 cells alone had hardly any impact on the rapidly growing ALL in control mice in our experimental conditions, they significantly inhibited leukemia burden and depleted normal B cells in IFN mice (Fig. 8b and Supplementary Fig. 14c. IFN + CART19 vs CTRL, IFN + CART19 vs CTRL + CART19, IFN + CART19 vs IFN, IFN vs CTRL, IFN vs CTRL + CART19 ****$p <$ 0.0001, Mantel–Haenszel test and Bonferroni correction). Reminiscent of the finding with OT-I T cells, CART19 cells revealed early and transient LAG3 and PD1 up-regulation in IFN but not control mice, reaching the highest peak in a long-term IFN survivor (Fig. 8c and Supplementary Fig 14d, IFN long-term survivor shown in green). Intriguingly, NGFR expression was also upregulated on CART19 cells of IFN mice, likely reflecting increased activity of the phosphoglycerate kinase promoter (PGK) in response to higher metabolic activation of CART cells in IFN mice (Supplementary Fig. 14e, f). Since in our LV NGFR and CAR19 expression are co-regulated by the PGK promoter, higher expression of the latter may have also favored more efficient CD19 + ALL killing in IFN mice. We also tested an improved CD19 CAR version (iCAR19), which contains inactivating mutations in the first and third CD3zeta ITAM domains and was reported to improve killing efficiency and increase T cell viability as compared to the standard CAR19[17]. We treated control or IFN mice with T-cells expressing either CAR19 or control T-cells at low or high leukemia burden (Fig. 8d and Supplementary Fig. 15a, b). Whereas CART19 and iCART19 cells had detectable but not significant effect on tumor burden in control mice, they significantly inhibited ALL in IFN mice in either early and late intervention trials (Fig. 8e. IFN + CART19 vs CTRL + CART19, IFN + iCART19 vs CTRL + iCART19, IFN + iCART19 late intervention vs CTRL + iCART19, ****$p <$ 0.0001, nonparametric rank-based method for longitudinal data in factorial experiments). We also confirmed early LAG3 up-regulation in both CART19 cells from IFN mice (Fig. 8f). Longitudinal analyses revealed peaks of iCART19 expansion in IFN mice concomitant to ALL growth inhibition or clearance, and early up-regulation of NGFR/CAR19 expression in IFN mice (Supplementary Fig. 15c, d). Overall, a significant fraction of IFN mice treated with CART19 cells were still alive at the latest follow-up (Fig. 8g. IFN + CART19 vs CTRL + i/CART19, IFN + iCART19 vs CTRL + i/CART19, IFN + iCART19 late intervention vs CTRL + i/CART19, ****$p \leq$ 0.0001, Mantel–Haenszel test and Bonferroni correction).

## Discussion

Here we show that gene-based delivery of IFNα by TIE2 + monocyte/macrophages in an ALL mouse model reprograms the leukemia-induced immunosuppressive TME towards effective priming and deployment of T-cell responses against multiple surrogate TSAs, attaining tumor clearance and protection from re-challenge.

Using immunophenotyping, bulk and single-cell transcriptome analyses, we provide a comprehensive characterization of a leukemia TME in a mouse model reproducing functional and transcriptional features of primary human B-ALL[14]. This integrated analysis points to non classical monocytes as major contributors to the immunosuppressive TME in leukemia, as these cells showed the most marked expansion and differential gene expression from the tumor-free condition among the myeloid populations studied. Of note, upregulation of IL-10 and downregulation of MHC-II genes were the most prominent changes consistent with the acquisition of immunosuppressive features. These alterations were effectively prevented by IFN gene therapy, which imposes an ISG and immune activation gene signature, resulting in a transcriptome closer to that observed in tumor-free mice. The described changes were relevant to the therapeutic benefits of gene therapy, as the extent of reprogramming observed at single-cell resolution correlated with leukemia inhibition. Although we did not further investigate it, we also observed an unexpected neutrophil heterogeneity, which was to some extent responsive to tumor and IFN induced changes in the TME.

IFN may affect several steps of the cancer immunity cycle[20], promoting priming, activation, expansion and persistence of OVA-specific T-cells, both endogenous and adoptively transferred. On the contrary, OVA-specific T-cells were hypofunctional and showed phenotypic evidence of exhaustion in control mice. The different response of OVA-specific T cells between IFN and control mice might be due to increased homing to the leukemia-infiltrated organs, more effective co-stimulation in the context of a pro-inflammatory microenvironment[21], increased expression and regulation of genes involved in CTL function (i.e., granzyme B, IFNg, Tbet, Eomes)[22], lower suppression from tumor and other infiltrating cells, and more favorable effector to target ratio because of IFN-mediated ALL growth inhibition. Of note, CAR19 expression increased on the infused T cells in IFN mice, likely in response to metabolic activation, further contributing to improved anti-tumor activity. In line with our results, it was recently reported that the CAR configuration displaying the highest tumoricidal activity and persistence was associated with concomitant activation of IRF7/ IFN-beta signaling pathway[23].

The OVA-ALL model allowed addressing the emerging hypothesis that dynamic immune reactivity against multiple neoantigens is required to achieve durable responses in cancer immunotherapy[24,25]. Indeed, most surviving mice showed

**Fig. 7** Combination of IFN gene therapy with CTLA4 blockade or adoptively transferred OT-I T cells improves survival. **a** Survival curves of OVA-ALL-injected IFN ($n = 14$), CTRL ($n = 14$), IFN+αCTLA4 ($n = 14$), and CTRL+αCTLA4 ($n = 15$) mice, *$p < 0.05$, ***$p < 0.001$, Mantel–Haenszel test, adjusted by Bonferroni method. **b** Percentage of OVA-specific T cells within CD8$^+$ T cells in the PB of mice from **a**. *$p < 0.05$, **$p < 0.01$, ****$p < 0.0001$, nonparametric rank-based method for longitudinal data in factorial experiments. **c** Clonality and similarity of the TCR-beta CDR repertoire of surviving mice (each arrow represents a mouse) from **a** before OVA-ALL injection (start of the arrow, $T_0$) and 4 days after second tumor challenge (tip of the arrow, $T_2$). **d** Survival curves of IFN + αCTLA4 ($n = 10$) and CTRL + αCTLA4 ($n = 12$) mice, *$p < 0.05$, Mantel–Haenszel test. **e** Survival curve of mice from **d** stratified based on their immune reactivity assessed by IFNγ-ELISPOT assay. Mice shown in red react against 2 or more surrogate tumor-specific antigens; mice shown in black react against 1 or no antigens. Note that reactivity against OVA and NGFR counts for 1 as their expression is co-regulated by a bidirectional promoter present within the lentiviral vector. **f, g** Experimental design (**f**) and OVA-ALL growth over time (**g**) (percentage in PB; mean ± SEM) in IFN ($n = 7$) or CTRL ($n = 7$) mice. **h** Absolute numbers (mean ± SEM) of OT-I T cells in BM and spleen of IFN ($n = 7$), CTRL ($n = 7$) and non tumor-bearing (C57Bl/6 no tumor, $n = 3$) mice 3 days upon OT-I adoptive transfer. *$p < 0.05$, **$p < 0.01$, Mann–Whitney. Each dot represents a mouse. **i** Experimental design (left) and survival curve (right) of OVA-ALL-injected CTRL ($n = 12$), IFN + OT-I ($n = 9$) and CTRL + OT-I ($n = 10$) mice. *$p < 0.05$, ****$p < 0.0001$ Mantel–Haenszel test, adjusted by Bonferroni test. **j** Absolute numbers (mean ± SEM) of adoptively transferred OT-I T cells in the PB over time of CTRL + OT-I and IFN + OT-I mice from **i**. Long-term surviving mice are shown in green. Each line represents a mouse. **k** Percentage (mean ± SEM) of Lag3 expression on OT-I T cells in the PB of mice from **i**. Long-term surviving mice are shown in green. Each line represents a mouse

spreading of the anti-tumor immune repertoire to encompass multiple surrogate TSAs, which conferred long-lasting protection against both OVA-expressing and immune-selected OVA-negative leukemia. Among these tumor-specific T cell clones, we identified those directed against the predicted surrogate TSA derived from our cell engineering strategy, but it is conceivable that additional T cell clones might have been generated against unknown tumor antigens.

The rapid in vivo induction of robust immunity against multiple surrogate TSAs by our strategy may have translational

implications for cancer immunotherapy, albeit with the caveat that TSAs commonly used in experimental cancer models are xenogeneic proteins and provide only a surrogate of clinically relevant neo-antigens, which are typically altered self-proteins carrying few amino acid substitutions and are thus likely to be less immunogenic. Because the latter neo-antigens arise as a consequence of patient-specific mutations, they are private and represent a challenge for current strategies aimed at generating protective immunity, which require tumor exome sequencing, prediction of immunogenic epitopes within the patient-specific

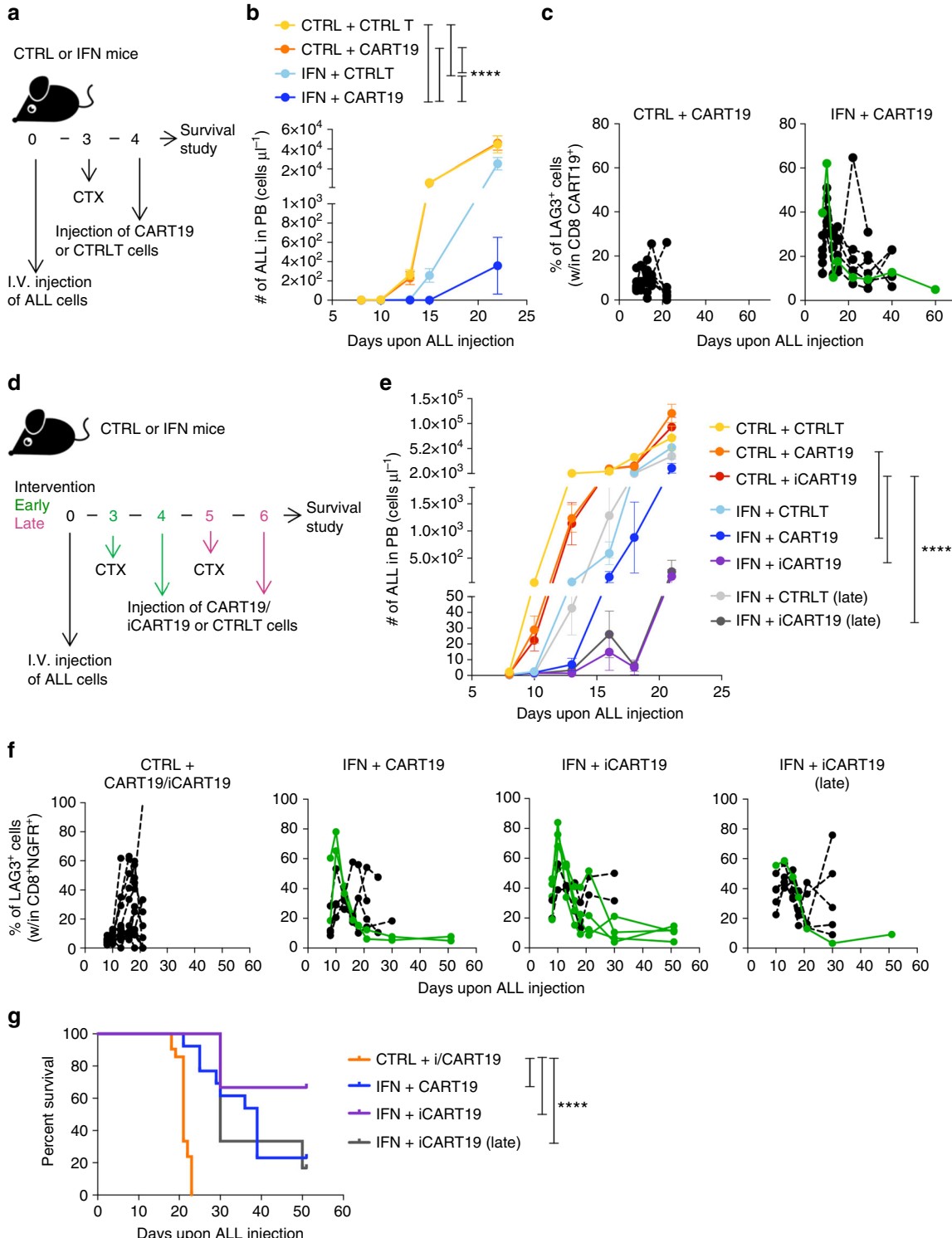

HLA, and generation of neo-antigen-specific T cells by ex vivo expansion, TCR gene transfer or vaccination[6,26–28]. On the contrary, if our approach were similarly efficacious against neo-antigens arising spontaneously in human tumors as shown here for experimental TSAs, it would have the advantages of not requiring knowledge of the neo-antigens to be targeted, and of targeting multiple TSAs at the same time, which should diminish the risk of immune evasion. We should also mention that, whereas our IFN gene therapy also inhibited the growth of the parental ALL, the introduction of a dominant TSA such as OVA might favor an initial cytotoxic response robust enough to allow effective spreading of the immune repertoire to multiple surrogate TSAs and establish durable protection. It is possible that the requirement for a strong TSA might reflect the very rapid course of the disease in the transplant setting and may not apply to spontaneous tumors arising in patients, where on the other hand the immune system is exposed to TSAs for a long time resulting in profound tolerance.

Other strategies have been developed to target IFNα to tumors by coupling the cytokine to tumor-specific antibodies, supporting the therapeutic potential of targeted IFN delivery in a broad range of tumors[29,30]. It is conceivable that a cell-based delivery strategy based on tumor-infiltrating macrophages might result in improved drug penetration and sustained bioavailability within the tumor. Of note, chronic exposure to type I and II IFNs has been reported to induce negative feedback mechanisms counteracting protective responses in infection and cancer[31–33]. We have not observed evidence of immunosuppressive effects of our IFN gene therapy, as shown by efficient clearance of viral infection in long-term transplanted mice[13] and protection from serial tumor challenges (this work), possibly because of the low-level sustained IFN expression induced in the target tissues. Furthermore, single-cell RNAseq analysis showed that IFN gene therapy, while effectively preventing ALL-driven upregulation of immunosuppressive genes in monocytes and macrophages, had limited impact on their transcriptome without tumor challenge (see trajectory in Fig. 5b left panel).

The composition of the immune cell infiltrate and the type of immune gene signatures are reliable predictors of clinical outcome in many cancers. An increased content of CD8 and memory T cells and Th1-skewed gene signatures have been linked to a more favorable prognosis, whereas increased intra-tumoral M2-like macrophages is associated with poorer prognosis[34,35]. Because our strategy can impose the immunological features associated with good clinical outcome, it prompts further development towards clinical translation. IFN gene therapy might provide long-term disease control, after remission has been induced by standard combination therapy, similarly to our mouse models where we prophylactically transplanted LV-IFN transduced HSC before tumor challenge.

The combination of IFN gene therapy with checkpoint blockade or adoptive T cell therapy substantially improved mouse survival also to a challenge with parental (OVA-negative) ALL. Whereas IFN enhanced T cell effector function, its pleiotropic effects on the TME may concomitantly enhance endogenous T cell priming against multiple weak TSAs[8,36] and contribute to the powerful synergy shown with these immunotherapies. It was recently reported that activation of the CD28/B7 pathway is required for effective rescue of CD8 T cells in PD-1 blockade therapy[37,38]. Thus, IFN gene therapy might enhance the efficacy of immunotherapy and broaden its reach to tumors with low mutational load or lacking dominant neo-antigens, and to solid tumors where T cell penetration and effector activity is often rate-limiting[39].

Hematological malignancies that foresee HSC transplantation as standard-of-care consolidation treatment[40] might provide a suitable framework for the first clinical testing of our IFN gene therapy strategy, as some HSC infused to recover from chemotherapy could be engineered for monocyte-mediated IFN delivery.

## Methods

**Experimental design**. Sample size was chosen according to previous experience with experimental models and assays. No sample or animal was excluded from the analyses. Mice were randomly assigned to each experimental group. Investigators were not blinded.

**Plasmid construction and LV production**. The m*Tie2*-IFN-mirT, m*Tie2*-GFP-mirT, PGK-OFP, PGK-tTA LV were previously described[12–14]. The NGFR-OVA transfer BdLV was generated by cloning the OVA cDNA (AgeI–SalI) amplified from the PGK-OVA LV[41] by PCR in place of the GFP cDNA (AgeI–SalI) in the NGFR-GFP BdLV[42]. The primers used were the followings: primer Fw: 5′-CGACCGGTCCACAAAGACAGCACCATGACA; primer Rv: 5′-ATTGTCGACTTAAGGGGAAACACATCTGCCAAAGA. The NGFR-CD20 transfer BdLV was generated by cloning the codon optimized human CD20 cDNA from a synthetized plasmid (KpnI blunt–SalI) in place of the GFP cDNA (AgeI blunt–SalI) in the NGFR-GFP BdLV. The NGFR-CAR19 and NGFR-iCAR19 transfer BdLVs were generated by cloning the CAR19 and inactive CAR19 sequences previously described[17] by PCR in place of the GFP cDNA (AgeI–SalI) in the NGFR-GFP BdLV using the following primers: primer Rv (inactive CAR19): 5′-AAACAGCTCCTCGAGTTATCTAGGGGCCA; primer Rv (CAR19): 5′-AAACAGCTCCCTCGAGTCATCTAGGGGCCAGT; primer Fw (CAR19 and inactive CAR19): 5′-AACACCGGTGTACCGAATTCATGGGCGTG. Concentrated VSV-G-pseudotyped LVs were produced and titered as previously described[43].

**Mice**. C57Bl/6 Ly45.2 and Ly45.1 mice of 6–8 weeks of age were purchased from Charles River Laboratory. C57Bl/6 Ly45.1/Ly45.2 were obtained by crossing C57Bl/6 Ly45.2 and C57Bl/6 Ly45.1 mice in the San Raffaele Scientific Institute animal research facility and used as donors for HSPC transplant. Transgenic OT-I C57Bl/6 Ly45.2 mice were maintained as colony at the San Raffaele Scientific Institute animal research facility. All animal procedures were performed according to protocols approved by the Animal Care and Use Committee of the San Raffaele Scientific Institute (IACUC 600, 836) and communicated to the Ministry of Health and local authorities according to the Italian law.

**Fig. 8** IFN gene therapy boosts activation of adoptively transferred CAR19-transduced T cells and enhances survival. **a, b** Experimental design (**a**) and ALL growth over time (**b**, absolute numbers in PB; mean ± SEM) in CTRL + CTRLT (*n* = 5), IFN + CTRLT (*n* = 5), CTRL + CART19 (*n* = 7), IFN + CART19 (*n* = 7). ****$p < 0.0001$, nonparametric rank-based method for longitudinal data in factorial experiments. **c** Percentage (mean ± SEM) of Lag3 expression on CD8 + NGFR + CART19 cells in the PB of mice from **b**. Long-term surviving mice are shown in green. Each line represents a mouse. **d, e** Experimental design (**d**) and ALL growth over time (**e**, absolute numbers in PB; mean ± SEM) in CTRL + CTRLT (*n* = 7), IFN + CTRLT (*n* = 6), CTRL + CART19 (*n* = 7), IFN + CART19 (*n* = 6), CTRL + iCART19 (*n* = 7), IFN + iCART19 (*n* = 6), IFN + CTRLT late (*n* = 6, late intervention trial), IFN + iCART19 late (*n* = 6, late intervention trial). ****$p < 0.0001$, nonparametric rank-based method for longitudinal data in factorial experiments. Statistical analysis is performed on selected groups (see Supplementary Tables 9 and 10). **f** Percentage overtime (mean ± SEM) of Lag3 expression on CD8 + NGFR + CART19 cells in the PB of mice from **b**. CTRL + CART19 and CTRL + iCART19 mice are plotted together. Long-term surviving mice are shown in green. Each line represents a mouse. **g** Survival curve of mice treated with CART19 or iCART19 cells from **b**, **e**. CTRL mice treated with CART19 and iCART19 from **b**, **e** are plotted together (CTRL + i/CART19, *n* = 21), IFN + CART19 (*n* = 13), IFN + iCART19 (*n* = 6), IFN + iCART19 late (*n* = 6). ****$p ≤ 0.0001$, Mantel–Haenszel test, adjusted by Bonferroni method

**Hematopoietic stem progenitor cell (HSPC) transplantation**. BM was harvested from male and female six-week-old C57Bl/6 mice and lineage-negative cells were purified by immuno-magnetic isolation (lineage cell depletion kit mouse, Miltenyi, #130–090–858). HSPC transduction, culture and transplantation in recipient female six-week-old C57Bl/6 mice was performed as previously described[12]. All animal procedures were performed according to protocols approved by the Animal Care and Use Committee of the San Raffaele Scientific Institute (IACUC 600, 836) and communicated to the Ministry of Health and local authorities according to the Italian law.

**Tumor studies**. The OVA-ALL sub-clone was generated by transducing the ALL from mouse #11[14] with the NGFR-OVA BdLV. Briefly, Briefly, ALL were kept in culture at a concentration of $2 \times 106$ cell/ml in Stem Span supplemented with 10% FBS, penicillin (100 IU/ml), streptomycin (100 µg/ml), 2% glutamine, IL3 (20 ng/ml), SCF (100 ng/ml), FLT3L (100 ng/ml), TPO (50 ng/ml) and transduced at $2 \times 10^7$ TU/ml with the NGFR-OVA BdLV. After 6 h of LV exposure, transduced ALL were washed and intravenously injected in sub-lethally irradiated (450 cGy) recipient C57Bl/6 Ly45.2 mice. Leukemic mice were sacrificed 14 days later, when ALL reached 90% of total PB cells, and BM was harvested by flushing the femurs and tibias. Transduced cells were isolated by immune-magnetic beads (CD271 MicroBead Kit, Miltenyi, #130-099-023). For tumor challenge mice were intravenously injected with $10^5$ OVA-ALL or $3 \times 10^4$ parental ALL. For re-challenge we mixed OVA-ALL and parental (OVA-negative) ALL at a 1:1 ratio ($10^5$ total ALL). For CTLA-4 blocking experiments 200 µg anti-CTLA4 (clone 9D9 BioXCell, #BE0164) or isotype control antibody (clone MCP-11 BioXCell, #BE0086) were administered intra-peritoneally at day 3 upon leukemia injection followed by 100 ug every 3–4 days for a total of five infusions. For CD8+ T cell depletion, 200 µg anti-CD8 depleting antibody (clone 53–6.72 BioXCell, #BE0004-1) was administered one day before OVA-ALL injection and then every 3 days.

**Adoptive OT-I T cell transfer**. For adoptive T cell experiments, OT-I T cells were purified from the spleen of 8 week-old transgenic female OT-I C57Bl/6 Ly45.2 mice by immune-magnetic selection (CD8+ T cell isolation kit, Miltenyi, #130-104-075). $1 \times 10^6$ naive OT-I T cells were intravenously injected in transplanted IFN or CTRL mice at the indicated time upon OVA-ALL injection.

**Generation of CART19 cells**. T cells were first purified from the spleen of 8 weeks-old female C57Bl/6 CD45.2 + mice by immune-magnetic selection (Pan T cell isolation kit, Miltenyi, # 130-095-130) and subsequently activated with anti-CD3/CD28 Dyna beads (ThermoFisher # 11452D), according to manufacturer's instructions. T cells were cultured in RPMI supplemented with 10%FBS, penicillin (100U/ml), streptomycin (100ug/ml), 1% glutamine, IL2 (30 u/ml), IL7 (5 ng/ml), IL15 (5 ng/ml), Na Pyruvate (1 mM), Hepes (20 mM), NEAA (1uM) and Beta-Mercaptoethanol (0.05 mM). One day after activation, T cells were transduced at a concentration of $1 \times 10^6$ cells/ml with $10^8$ TU/ml of the NGFR-CAR19 or NGFR-iCAR19 BdLVs. 12 h after LV exposure T cells were washed and expanded in culture for 8 days prior to infusion in mice at a dose of $7 \times 10^6$ (experiment from Fig. 6b) and $10^7$ (experiment from Fig. 6e) NGFR + T cells. Mice are conditioned with 100 µg/g of cyclophosphamide prior to T cell infusion.

**In vivo proliferation assay**. In vivo proliferation assays were performed using 5-ethynyl-2′deoxyuridine (EdU, Invitrogen), a thymidine analog used in alternative to BrdU. Mice were injected i.p. with 100 µg of EdU 24 h before analysis. BM and spleen were harvested and processed according to manufacturer's instructions (Click-iT® EdU Flow Cytometry Assay Kit, Invitrogen, #C10636), and percentages of EdU incorporation into leukemic cell were measured by flow cytometric analysis.

**Flow cytometry**. All cytometric analyses were performed using the FACSCanto II and LSRFortessa instruments (BD Bioscience) and analyzed with the FlowJo software (v. 9.3, Tree Star Inc.).

Peripheral Blood (PB): for immunostaining a known volume of whole blood (100 µl) was first incubated with anti-mouse FcγIII/II receptor (Cd16/Cd32) blocking antibodies for 10 min at room temperature and then incubated in the presence of monoclonal antibodies (for antibodies see Supplementary Table 1) for 15 min at room temperature. Erythrocytes were removed by lyses with the TQ-Prep workstation (Beckman-Coulter) in the presence of an equal volume of FBS (100 µl) to protect white blood cells. For quantitative flow cytometry we used Flow-count Fluorospheres (Beckman Coulter, #7547053), according to manufacturer's instruction. For OVA-specific pentamer staining, whole blood was first lysed with $H_2O$ and $1 \times 10^6$ PBMCs were then stained in 50 µl of PBS containing 2 mM EDTA and 0.5% bovine serum albumine (BSA) according to manufacturer's instructions. To exclude dead cells from the analysis, cells were washed and resuspended in PBS containing 2% FBS and 10 ng/ml of 7-aminoactinomycin D (7-AAD). For Treg cells staining we first performed surface staining on 50 µl of whole blood (for antibodies see Supplementary Table 1) and then blood was fixed and permeabilized (eBioscience, #00-5523-00) for 20 min at room temperature, washed and resuspended in 50 µl of Permeabilization solution containing the anti-mouse FoxP3 or isotype control antibodies and incubated for 20 min at room temperature.

Bone marrow (BM) and spleen: BM cells were obtained by flushing the femurs in PBS 2% FBS solution and by passing cell suspension through a 40 µm nylon filter. Spleens were first smashed and lysed by $H_20$ to remove erythrocytes. The obtained cells suspension was passed through 40 µm nylon filter and washed in cold PBS containing 2 mM EDTA and 0.5% BSA. For immunostaining cells ($1 \times 10^6$–$3 \times 10^6$ cells) were resuspended in 100 µl of PBS containing 2 mM EDTA and 0.5% bovine BSA, and incubated with anti-mouse FcγIII/II receptor (Cd16/Cd32) blocking antibodies for 15 min at 4 °C. Staining was then performed with monoclonal antibodies (for antibodies see Supplementary Table 1) for 20 min at 4 °C. OVA-specific pentamer staining and Treg staining was performed on $1 \times 10^6$ cells, as described above for PB. For all intracellular staining and for some surface staining the LIVE/DEAD Fixable Dead Cell Staining (ThermoFisher, #L34959) was used to discriminate alive and dead cells according to manufacturer's instructions. Cell cycle and cell apoptosis staining on BM cells and splenocytes from OVA-ALL injected mice were performed as previously described[14].

**ELISPOT assay**. The γIFN ELISPOT was performed in 96-well flat-bottomed plate (Millipore) coated with 5 µg/ml of anti-mouse IFN-γ primary antibody (BD, #554430). Splenic CD8+ effector T cells were first purified by immune-magnetic cell labeling (CD8 MicroBead Kit, Miltenyi, #130-049-401). Effector T cells were subsequently plated at a final concentration of $10^5$ or $2 \times 10^5$ cells/well in presence of irradiated (6000 cGy) $10^5$ EL4 target cells. For some experiments, effector T cells were stimulated with 2.5 µg/ml of Concanavalin A, as control. When using total PBMCs as effector cells, total blood was first lysed with $H_2O$ to eliminate erythrocytes, recovered PBMCs were plated at a final concentration of $10^5$ cells/well. After 42 h of incubation at 37 °C/5%CO2, γIFN-specific spots were revealed as previously described[44]. γIFN-spots were quantified with ELI-Expert.Elispot-Reader and analyzed by Eli.Analyze Version 5.1 (A.EL.VIS).

**RNA extraction, qPCR and gene expression analysis**. RNA extraction was performed using the RNeasy Plus mini Kit (Qiagen) according to manufacturer's instructions and retro-transcribed using the SuperScript Vilo kit (11754250; Invitrogen). All Q-PCR analyses on single genes were done using TaqMan probes from Applied Biosystems (see below). Q-PCR was run for 40 cycles using the Viia 7 instrument and raw data (Ct) were analyzed as previously described[12]. The following Taqman probes were used on purified splenic mouse CD4+ T cells: Il10 (Mm00439616_m1), Il17 (Mm00439618_m1), Ifn-γ (Mm01168134_m1), Il2 (Mm00434256_m1), Tbx21 (TBET) (Mm00450960_m1), Rorc (Mm01261022_m1), Foxp3 (Mm00475156_m1), Gata3 (Mm00484683_m1), Il22 (Mm00444241_m1), Il4 (Mm00445259_m1).

**TCR sequencing**. Input DNA was obtained from PBMCs of IFN and CTRL mice before tumor injection ($T_0$), 30 days upon OVA-ALL injection in tumor-free long-term surviving mice ($T_1$) and 4 days upon re-challenge with OVA-ALL and parental ALL mixed at 1:1 ratio ($T_2$). TCRβ chain sequencing was performed at Adaptive Biotechnologies using the ImmunoSEQ platform with primers specific for all 54 known expressed Vβ and all 13 Jβ regions. Each unique CDR template at the aminoacid level was quantified in counts per million (cpm). Clonality was evaluated as 1 – Pielou's evenness[45]:

$$C = 1 - \frac{H'}{H}$$

where H' is the entropy of a sample, calculated on template counts higher than 0 (i.e., the set of all observable template in a mouse) and H is the maximal theoretical entropy for a mouse, defined by

$$H = \ln(S)$$

where S is the number of distinct templates in a mouse.

Similarity was evaluated as the 1−Bray–Curtis distance between a sample s and the mean of template counts at time t.

$$S_t = 1 - \frac{\sum_i |s_{i,t} - \overline{s_{i,t}}|}{\sum_i |s_{i,t} + \overline{s_{i,t}}|}$$

This measure is equivalent to Sørensen-Dice similarity index and it was evaluated including all i observed templates at time t.

**Bulk RNA-sequencing**. RNA was extracted from 10,000 to 50,000 sorted cells using the ReliaPrep RNA MiniPrep System (Promega) and RNA-Seq libraries were prepared with the SMART-Seq2 protocol (PMID 24385147), with minor modification. Briefly, RNA (1–5 ng) was reverse transcribed using custom oligodT and template-switching LNA oligos (sequences), followed by PCR amplification and clean-up (Ampure XP beads, Beckman Coulter). The resulting cDNA (0.5–1 ng) was tagmented at 55 °C for 30 min and final RNA-Seq libraries generated using reagents from the Nextera XT DNA Library Prep Kit (Illumina). Sequencing was performed on a NextSeq 500 machine (Illumina, San Diego, CA) using the NextSeq 500/550 High Output v2 kit (75 cycles).

**Single-cell RNA-sequencing.** Droplet-based digital 3′ end scRNA-Seq was performed on a Chromium Single-Cell Controller (10 × Genomics, Pleasanton, CA) using the Chromium Single Cell 3′ Reagent Kit v2 according to the manufacturer's instructions. Briefly, suspended single cells were partitioned in Gel Beads in Emulsion (GEMs) and lysed, followed by RNA barcoding, reverse transcription and PCR amplification (12–14 cycles). Sequencing-ready scRNA-Seq were prepared according to the manufacturer's instructions, checked and quantified on 2100 Bioanalyzer (Agilent Genomics, Santa Clara, CA) and Qubit 3.0 (Invitrogen, Carlsbad, CA) instruments. Sequenced was performed on a NextSeq 500 machine (Illumina, San Diego, CA) using the NextSeq 500/550 High Output v2 kit (75 cycles).

**RNA sequencing analyses.** Reads were generated on NextSeq 500 (illumina) instrument following manufacturer recommendations. Single end reads (75 bp) were aligned to the mm10 reference genome using STAR aligner[46]. FeatureCounts function from Rsubread package (v 1.16)[47] was used to compute reads over RefSeq *Mus musculus* transcriptome, with option minMQS set to 3. Only genes with a CPM (Counts per million) value higher than 1 in at least two samples were retained. Coefficient of determination ($R^2$) was computed for each couple of samples on log transformed RPKM (Reads per kilobase per million) values of expressed genes. Further analyses were performed with edgeR R package (v 3.8.6)[48]. Read counts were normalized with the Trimmed Mean of *M*-values (TMM) method[49] using calcNormFactors function and dispersion was estimated with the estimateDisp function. Differential expression across different conditions was then evaluated fitting a linear model on the dataset with glmFit and glmLRT functions. Differentially expressed genes (DEGs) for each comparison were defined setting a cutoff of 0.01 on FDR and retaining genes with an absolute logFC higher then 1.5. Gene Ontology enrichment analysis was then performed on DEGs. Enriched Gene Ontology terms ($p < 1e-3$) were calculated using GOrilla[50]. The enrichment was calculated against the background of expressed genes.

**Single cell RNA sequencing analyses.** Data processing: Reads for the single cell experiments were generated on NextSeq 500 (illumina) instrument following manufacturer recommendations.

Fastq files were processed with Cell Ranger (v. 1.3, https://support.10xgenomics.com/single-cell-gene-expression/software/pipelines/latest/what-is-cell-ranger) using default parameters. Reads were aligned to reference genome mm10; genes were quantified using ENSEMBL genes as gene model. UMI and cell barcodes were then filtered as described in ref. Only confidently mapped reads, non-PCR duplicates, with valid barcodes and UMIs were retained to compute a gene expression matrix containing the number of UMI for every cell and gene.

Gene counts were imported in R environment (v. 3.3.2) and processed with Seurat (v 2.1, http://satijalab.org/seurat/). Cells expressing less than 300 unique genes were discarded. Counts were normalized using Seurat function NormalizeData with default parameters. Genes with a mean expression lower than 0.01 were excluded. Cells with a ratio of mitochondrial versus endogenous genes expression exceeding 0.1 were also excluded. Expression data were than scaled using ScaleData function, regressing on number of UMI, percentage of mitochondrial gene expression and difference between S and G2M scores. Cell cycle scores were calculated using CellCycleScoring function.

Graph-based clustering: Most variable genes across the dataset were identified using FindVariableGenes function; genes with average expression lower than 0.01 and higher than 3 were excluded and a cutoff of 0.5 was applied over dispersion z-scores. The first 25 principal components were then evaluated on the resulting 1031 genes. Cell clusters were then defined at resolution $r = 0.23$ using FindCluster function. Cells were visualized in 2-dimension using t-SNE (t-Distributed Stochastic Neighbour Embedding).

Genes enriched in cells within each cluster were identified. For each cluster $K$ the log-fold change of expression for each gene was calculated as

$$C_{i,j} = e^{N_{i,j}} - 1$$

$$\mathrm{logFC}_i = \log_2\left(\frac{\sum_{j \in K} C_{i,j}}{n_K}\right) - \log_2\left(\frac{\sum_{j \notin K} C_{i,j}}{n_{\mathrm{tot}} - n_K}\right)$$

with $j$ being the cell, $i$ the gene and $N$ the log-normalized expression value.

For each cluster genes expressed in at least 20% of cluster single cells were selected and ranked by decreasing logFC.

Gene signatures for different cell type populations: We compiled several lists of genes defining different lineages in blood. Briefly, we downloaded expression data from Immgen RNA-Seq dataset (https://www.immgen.org/). For each cell type (Neutrophils, Dendritic cells, NK cells, T cells and B cells) we extracted genes being differential in all pairwise comparisons with other cell types using glmFit and glmLRT functions from edgeR R package as described above. For each comparison genes were called significant at FDR < 1e−4 and the top 200 ranked for decreasing fold change were selected. The final gene signature defining each cell type was the intersection of the resulting 4 lists.

As monocytes were not included in Immgen dataset, we downloaded monocytes data from[51] and calculated DEGs with edgeR as above. We retained the top 5

genes, ranked by decreasing fold change, defining classical and non-classical Monocytes.

Macrophages included in Immgen dataset were from peritoneal cavity, hence we used signature defined in[52] for macrophages from spleen.

Analysis of non-classical monocytes (cluster1): Genes differentially expressed across different condition in cluster1 were identified computing a logFC for each gene with the procedure described before. The top 100 genes upregulated and downregulated in ALL versus CTRL and IFN + ALL versus ALL comparisons were identified ranking genes on logFC values. Only genes expressed in at least 5% of cells in at least one sample were considered. Gene Ontology enrichment analysis was then performed on these sets of genes. Enriched Gene Ontology terms ($p < 1e-3$) were calculated using GOrilla[50]. The enrichment was calculated against the background of expressed genes. The subset of cells included in cluster 1 (defined above) was then re-clustered separately with the before mentioned procedure. Clusters were defined at resolution $r = 0.5$. Minimum Spanning Tree was calculated adapting exprmclust function from TSCAN[53] on Seurat data structure.

**Statistical analysis.** Values are expressed as mean ± standard error of the mean (SEM) as indicated. Statistical analyses were performed by Mann-Whitney test or Kruskall–Wallis test followed by Dunn post-test correction, unless otherwise indicated. Differences were considered statistically significant at $p < 0.05$. Analyses were performed in R 3.2.2[54], NPC Test R10[51] and Prism (GraphPad Prism version 7.0). Analysis of cell populations in Supplementary Fig.12c was performed by nonparametric combination test[55], which is a permutation-based alternative test to the parametric Hotelling *T*-square two-sample test. This methodology allows performing multivariate comparison of the means in groups with small number of observations which do not satisfy the assumption for multivariate normality. Figures 1c, d, h, 7b and 8b, e were modeled within a nonparametric framework which allows accounting for small sample size, the presence of outliers, non-Gaussian and heavily skewed data distribution, for which parametric procedures are not appropriate. A robust and flexible rank-based method suitable for the longitudinal analysis in factorial design was applied (see Supplementary Tables 2–4, 6, 8–10)[56,57]. In this context hypothesis testing is focused on detecting differences in the distributions of collected variables rather than on difference between means[58,59]. This analysis was implemented by using nparLD package developed in R[58]. Survival curves in Fig. 6a, Fig. 7a, d, j, Fig. 8g and Supplementary Fig. 1a were estimated by means of Kaplan-Meier (KM). The event variable considered here was time to death, no informative censoring occurs. Nonparametric log-rank (Mantel–Haenszel) test statistics were used to compare the $k$ survival curves related to the $k$-samples. In the presence of more than 2 groups, multiple comparison issue was addressed by adjusting $p$-value by the Bonferroni method (see Supplementary Table 5, 7, 11). Survival analysis was performed using the R package survival[60].

**Data availability.** The RNA-seq data on sorted splenic macrophages have been deposited in the ArrayExpress database under the accession code E-MTAB-6482. The single-cell RNA-seq data on splenic CD11b + cells have been deposited in the ArrayExpress database under the accession code E-MTAB-6487.

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

## Acknowledgements

We thank A. Mondino and members of the Naldini lab for fruitful discussion, and the staff of the OSR genomic (CTGB) and cell sorting (FRACTAL) facilities; A. Pramov, for help with statistical analyses and C. di Serio, who supervised A. Pramov and C.B. This work was supported by grants from the EU (ERC Advanced Grant 249845 TARGETINGGEN-ETHERAPY) and the Italian Association for Cancer Research (AIRC) to L.N., an EHA Clinical Research Fellowship to B.G., and an ERC Starting Grant (X-TAM) to R.O.

## Author contributions

G.E. designed, performed, analyzed experiments, and wrote the manuscript. L.B. performed and analyzed experiments. G.B. performed bioinformatics analyses of scRNA-seq and RNA-seq data, M.N. performed CART19 experiments, M.G. performed scRNA-seq experiments, A.R. and T.P. performed experiments. B.C. performed CART19 experiments, D.C. performed bioinformatics analysis of TCR sequencing data, C.B. performed statistical analysis, A.A. provided intellectual input and edited the manuscript. A.B. provided intellectual input and supervised M.N. and B.C., R.O. provided intellectual input, supervised M.G. and G.B, and wrote the manuscript, B.G. and L.N. designed and coordinated the research and wrote the manuscript. G.E. conducted this study as partial fulfillment of her PhD in molecular and cellular biology at San Raffaele University, Milan, Italy.

## Additional information

**Competing interests:** L.N. and B.G. are inventors of patents filed by the San Raffaele Scientific Institute and Telethon Foundation related to technologies described in the present manuscript and own equity on Genenta Science, a biotechnology startup aimed at developing IFN gene therapy by tumor-infiltrating monocytes (www.genenta.com). The remaining authors declare no competing interests.

