## [Peer Review File · Nature Communications]

Reviewers' comments:

Reviewer #1 (Remarks to the Author):

The authors analyzed various combinations of immunotherapy for the treatment of B-ALL, combining two experimental model systems published previously by the group. It uses an acute B cell leukemia transplantation model in bone marrow chimeric mice expressing IFN α from the Tie2-promotor. Supposedly local IFN α , CTLA4 blockade or adoptive T cell therapy were tested alone or in combination. The T cells response to different xenogeneic proteins (OVA, rtTA and OFP, truncated h-NGFR) expressed by the B-ALL as surrogate tumor antigens was investigated. IFN α slightly delayed outgrowth of B-ALL. CTLA4 blockade and IFN α yielded additive effects. T cell responses towards surrogate antigens artificially expressed by the leukemic cells were enhanced by IFN α . Transferred naïve OT-I T cells showed reduced exhaustion markers in the presence of IFN α .

General comments:

1. The study lacks novelty. All three immunotherapeutic interventions have been extensively published by the authors and others before. Randomly combining currently popular treatment regimens is not a great step forward.
2. The model is highly artificial. The title and abstract promise that neoantigens were investigated. However, the authors analyzed only xenogeneic proteins, artificially expressed in the leukemic cells. Clinically relevant neoantigens are typically altered self-proteins carrying a single amino acid substitution. B-ALL in the clinic certainly does not express antigens as immunogenic as entire xenogenic proteins.
3. A countless number of publications investigated ovalbumin as surrogate tumor antigens with similar or even better therapeutic effects. Ovalbumin as surrogate tumor antigen has also been analyzed in leukemia models (Zhou F., Rouse B.T., and Huang L. Cancer Res. 1992; Gerbitz et al. PLOS one 2012). In other studies, OT-1 T cell transfer alone was effective to reject large tumors, if ovalbumin was expressed in sufficient amounts (Engels et al, Cancer Cell 2013). Unclear how much ovalbumin is expressed in the current study.
4. A large number of publications by Belardelli over the last 20 years investigated the effect of local IFN α on tumor growth. Compared to these studies, the current study adds little except expressing IFN α from tie2-positive rather than tumor cells and this was published before.
5. The model is also artificial because treatment was started day 5 after tumor cell injection. The extremely fast growth kinetics, referred to by the authors as "fast forward model", probably leads very quickly to extremely high antigen amounts of foreign proteins and does not reflect the clinical situation the authors suggest to mimic.
6. The gene signature experiments are descriptive. The role of M1/M2 macrophages, NK cells or NKG2D remains unclear.
7. It is unclear where and how much IFN α is produced.
8. Countless numbers of publications have shown that mice, which rejected a cancer cell inoculum, are subsequently protected from a second challenge.

Minor Points:

The discussion is rather long.

ATT is usually not performed with naïve but effector T cells. What happens in their IFN α model if effector OT-I are transferred?

Discussion line 214, "neo-antigen that help to drive the transformed phenotype" is misleading since the driver is the mir-126 RNA and not one of the xenogeneic tumor antigens.

Supplementary Figure 4b lacks open symbols.

Reviewer #2 (Remarks to the Author):

The authors utilize a gene transfer technique to illustrate enhanced anti-tumor efficacy in the context of IFN α and is further enhanced by CTLA-4 blockade. The authors, utilizing OVA modified ALL tumor cells, show enhanced anti-tumor efficacy in the context of IFN α (through genetically engineered macrophages and monocytes) mediated by both endogenous T cells (primarily reactive to OVA) as well as OVA specific OT-1 T cells. The authors further demonstrate IFN mediated changes within the tumor microenvironment with increased M1 macrophages and Th1 responses. Utilizing this ALL model, the authors further demonstrate epitope spreading in the context of CTLA-4 blockade, a phenomenon which apparently is predictive of tumor eradication. Collectively, the authors argue that a gene therapy approach with IFN α may have clinical application in part allowing for recruitment and activation of T cells specific to patient specific tumor neo-antigens.

Critiques

1. Targeted delivery of pro-inflammatory cytokines is not a novel concept (see IL-12 modified T cells in the context of pmel studies and melanoma targeted TILs). In particular, IL12 studies similarly demonstrated changes in the tumor microenvironment by locally delivered pro-inflammatory cytokines.
2. There is some concern about the highly artificial nature of this immune competent tumor model using a non-self OVA antigen which is highly immunogenic and fails to reflect spontaneous tumors in the clinical setting. Thus, despite the improved outcomes seen in these studies, the highly artificial nature of the tumor makes it questionable whether this data would translate well to the clinic.
3. Gene expression profile data is confusing and almost can be considered raw data. These plots should be removed and replaced by a more interpretable and more fully analyzed summary.
4. In tumor re-challenge studies (figure 4a-d), it would be of great interest if the authors could study the T cells in greater depth (i.e. more directly demonstrate antigen spreading) and even study whether adoptive transfer of T cells from these mice are able to eradicate both OVA+ and OVA- tumors in tumor naïve mice. Further, in the text the authors point out that these mice were resistant to a challenge of a mix of OVA+ and OVA- tumor but this data is not provided.
5. With respect to studies presented in figure 4h, the authors again note that these mice, when re-challenged a mix of OVA positive and negative tumors, a majority of mice survived. Again, this data is curiously not provided.
6. Was statistical analysis conducted on the data presented in figure 4j-k? If not, why not?
7. There was an excessive amount of relevant data presented in the supplementary data making review of the manuscript difficult.
8. It is not clear how the authors propose to translate this approach to the clinical setting. Do the authors propose that patients undergo an autologous bone marrow transplant with IFN α modified stem cells after initial chemotherapy? I am not sure this is a clinically feasible approach for a vast majority of patients. This needs to be better clarified.

Reviewer #3 (Remarks to the Author):

Escobar et al present data showing that genetically driven gene-based IFN α delivery in a B-cell ALL model inhibits tumor growth and modulates the tumor microenvironment resulting in a more effective antitumor immune response. They suggest that combining this approach with CTLA4 blockade or adoptive transfer of tumor antigen specific T-cells results in improved survival in a murine model.

Specific comments -

1. A primary point of the manuscript is that the interferon gene delivery is targeted and produced by tumor infiltrating monocytes and macrophages. Previous models using this approach have been in solid tumors (glioma, breast, colorectal cancers) and not hematological malignancies which have continuous exposure to the peripheral circulation. IFN α has been used as therapy in various hematological malignancies with clinical benefit when administered systemically. It would be important to show that similar results would not be seen if interferon alpha was simply systemically administered. Furthermore, the authors suggest that this approach could be developed clinically. To do that, it would be important to show that IFN α transfected monocytes can be infused resulting in similar effects.

2. The manuscript highlights the importance of effector T-cells and suggests that the IFN α delivery modulates the microenvironment to promote T-cell function. The OVA-ALL model used does not seem to be very immunogenic and it would appear that OFP and tTA that are also expressed as neo-antigens after transfection of the cells may also be responsible for CD8+ cell activation. The authors should clarify which of these antigens is the most important. Furthermore, when CD8+ cells are depleted from the model, there is only a modest decrease in tumor burden suggesting that other mechanisms could be important. They suggest a slower proliferation rate of the tumor cells due to IFN α could be the cause. This should be confirmed. They further state that the delay in tumor cell proliferation may allow for the expansion of tumor-specific CTL but provide no data to support this. This should be done.

3. The authors report that OVA-specific T-cells upregulate LAG-3 and acquire a memory phenotype. When studied at the time of sacrifice, cells in IFN mice had downregulated PD-1 and LAG-3 but maintained the memory phenotype. They suggest that IFN α exposure prevented T-cell exhaustion. Given that PD-1 expression is also typically seen when cells are activated, it would be important to confirm that IFN α is protective against T-cell exhaustion by treating control OVA-specific T-cells ex-vivo and showing the same thing.

4. To enhance the IFN effect, the authors combined tumor targeted IFN α treatment with CTLA4 blockade. They should clarify why an anti-CTLA4 approach was chosen when their data suggests a role for PD-1 or LAG-3 blockade as upregulation of LAG-3 and PD-1 were seen in hypofunctional effector cells.

POINT BY POINT REPLY TO EACH REVIEWER

Reviewer #1 (Remarks to the Author):

The authors analyzed various combinations of immunotherapy for the treatment of B-ALL, combining two experimental model systems published previously by the group. It uses an acute B cell leukemia transplantation model in bone marrow chimeric mice expressing IFN α from the Tie2-promotor. Supposedly local IFN α , CTLA4 blockade or adoptive T cell therapy were tested alone or in combination. The T cells response to different xenogeneic proteins (OVA, rtTA and GFP, truncated h-NGFR) expressed by the B-ALL as surrogate tumor antigens was investigated. IFN α slightly delayed outgrowth of B-ALL. CTLA4 blockade and IFN α yielded additive effects. T cell responses towards surrogate antigens artificially expressed by the leukemic cells were enhanced by IFN α . Transferred naïve OT-I T cells showed reduced exhaustion markers in the presence of IFN α .

General comments:

1. The study lacks novelty. All three immunotherapeutic interventions have been extensively published by the authors and others before. Randomly combining currently popular treatment regimens is not a great step forward.

We respectfully disagree with this statement of the reviewer. The scope of our work was to develop a scientific rationale to translate IFN gene/cell therapy to the clinic in patients suffering from lymphoid malignancies. As evident from the detailed description of our work presented in the manuscript, our experimental plan was purposely and carefully designed and it is in no way the result of a random combination of popular treatments. The demonstration that our B-ALL model induces prominent changes in the orthotopic immune microenvironment towards an immune paralytic state, and that targeted IFN gene therapy reverses these changes was far from obvious and provides highly relevant information for moving towards a first-in-human clinical testing. Importantly, we have now generated a substantial amount of new data providing novel insights on the activity of myeloid cell-based delivery of IFN into the leukemia microenvironment (**new Fig. 3, Supplementary Fig.5,6,7,8, Supplementary Online Excel Table 4,5,6**, further discussed in the reply to the Reviewer's comment 6). Moreover, we introduced a novel, clinically-relevant model of T cell-based immunotherapy reproducing the findings we previously obtained with the OT-1 model, i.e. a remarkable increase in efficacy mediated by our IFN delivery strategy, using CAR19 T cells against the parental B- ALL model (shown in the **new Fig.6 and Supplementary Fig.14,15** and further discussed in the reply to the Reviewer's comment 3).

2. The model is highly artificial. The title and abstract promise that neoantigens were investigated. However, the authors analyzed only xenogeneic proteins, artificially expressed in the leukemic cells. Clinically relevant neoantigens are typically altered self-proteins

carrying a single amino acid substitution. B-ALL in the clinic certainly does not express antigens as immunogenic as entire xenogenic proteins.

We agree with the Reviewer that TAAs commonly used in experimental cancer models are xenogeneic proteins and provide only a surrogate of clinically relevant neo-antigens, which are typically altered self-proteins carrying few amino acid substitutions and are thus likely to be less immunogenic. We now explicitly mention this limitation in the discussion (lines 343-347) and have replaced the word neo-antigens with the more appropriate term “tumor associated antigen (TAA)”. However, we also show that IFN-gene therapy can inhibit the growth of the parental ALL, which lack the immune-dominant OVA antigen - although retaining the expression of other xenogeneic proteins instrumental to the generation of the tumor model (now Fig. 1a). Importantly, whereas CTLA4-blockade therapy failed to show any significant effect in control mice injected with the parental ALL, it significantly inhibited leukemia growth and improved mice survival when combined with IFN gene therapy (Fig 1c and Supplementary Fig.1a). These results indicate that our strategy can promote the induction of anti-tumor responses against the parental OVA-negative ALL cells, an outcome that can be further enhanced when combined with blockade of negative immune-checkpoints. Thus, we believe that the rapid in vivo induction of anti-tumor immunity directed against multiple surrogate TAA we observed in mice treated with IFN gene therapy may still be relevant from a translational standpoint, particularly if one considers that it occurs in a fast-growing ALL model.

3. A countless number of publications investigated ovalbumin as surrogate tumor antigens with similar or even better therapeutic effects. Ovalbumin as surrogate tumor antigen has also been analyzed in leukemia models (Zhou F., Rouse B.T., and Huang L. Cancer Res. 1992; Gerbitz et al. PLOS one 2012). In other studies, OT-1 T cell transfer alone was effective to reject large tumors, if ovalbumin was expressed in sufficient amounts (Engels et al, Cancer Cell 2013). Unclear how much ovalbumin is expressed in the current study.

We are aware that many other publications have used ovalbumin as surrogate tumor antigen, which in our opinion supports the value of such a model, and that adoptive transfer of OT-I T cells alone resulted in effective tumor rejection in other tumor models. Nonetheless, this was not the case in our study as OT-I T cell transfer in control mice resulted in only 20% long-term survival (now Fig 5j). Similarly, CTLA4-blockade therapy failed to show any significant effect in control mice injected with the parental ALL (Fig.1c and Supplementary Fig.1a) and only slightly improved the survival of mice injected with the OVA-ALL cells (Fig. 5a,d). These results may be due to the fast-growing features of our ALL model and, more importantly, due to the immunosuppressive microenvironment it induces as evidenced e.g. by upregulation of IL-10 and down-regulation of MHC-II genes, as we now demonstrate by bulk and single cell RNA seq (**new Fig. 3, Supplementary Fig.5,6,7,8, Supplementary Online Table 4,5,6**; see also response to point 6). Notwithstanding, when IFN gene therapy was combined with these immunotherapy strategies, durable responses increased up to nearly **70%** in the OT-I + IFN (Fig. 5j) and nearly up to **40%** in the anti-CTLA4+IFN mice injected with OVA-ALL cells (Fig. 5a,d) and tumor growth was significantly

delayed and survival improved in CTLA4+IFN mice injected with the parental OVA-negative ALL cells (Fig.1c and Supplementary Fig.1a).

The Reviewer seems also to suggest that the sub-optimal OT-I response in control mice could be due to insufficient expression of the Ovalbumin protein on leukemic cells. In order to address this point, we stained ALL cells with an antibody specific for the OVA_{SIINFEKL}-MHC-I complex. Our results, provided in the figure below showed that the immune-dominant OVA peptide was expressed at high levels on the surface of leukemic cells in the BM of IFN and control mice (grey histograms are the FMO controls). No expression of the OVA peptide-MHCI complex was observed on control CD11b+ myeloid cells. We don't think these data need to be included in the manuscript, given the obvious efficacy of anti-OVA responses induced in IFN gene therapy mice, but if the Editor would prefer otherwise we could add them in a supplementary figure.

To further address the Reviewer's concern regarding the use of OVA in our model we have now investigated whether IFN gene therapy could increase the anti-leukemic efficacy of T cells expressing a chimeric antigen receptor against murine CD19 (CART19), which is naturally expressed by these types of tumors. As shown in the **new Fig.6 and Supplementary Fig.14,15**, we found only a modest activity of this CART19 in control animals, at least in our experimental conditions, probably due to a) the aggressive nature of our parental (OVA-negative) B-ALL model, b) a relatively low degree of T lympho-depletion following cyclophosphamide conditioning in the mouse (as opposed to humans), and c) an ex vivo T-cell engineering protocol not fully optimized for mouse cells. Importantly, under the same conditions, IFN gene therapy-treated mice demonstrated substantial activity leading to the cure in up to two thirds of the mice. CART19 cells in responders showed transient (as opposed to persistent) Lag3 upregulation and signs of metabolic activation, providing new evidence that our IFN delivery strategy renders anti-tumor T cell effector activity more proficient. These results are in our view highly relevant, as they have been obtained without introducing nominal antigens in the leukemia, and they stringently model the current gold standard immunotherapy of B-ALL, based on CART19 treatment. In addition, interventions increasing CART cell efficacy may well have broader clinical implications, particularly for

new antigen targets and in solid cancer, where CART therapy has been less successful than in B- ALL.

4. A large number of publications by Belardelli over the last 20 years investigated the effect of local IFN α on tumor growth. Compared to these studies, the current study adds little except expressing IFN α from tie2-positive rather than tumor cells and this was published before.

We previously reported the toxicity observed after injecting intravenously a lentiviral vector expressing IFN α from an ubiquitously although moderately expressed promoter (PGK) in mice. This gene delivery strategy resulted in transduction of spleen and liver cells, which released high levels of IFN α in the serum (418 ± 124 pg/ml) and caused myelotoxicity, weight loss, thrombocytopenia and limited effect on the growth of a solid tumor. Conversely, IFN α levels were undetectable in TIE2-IFN transplanted mice and resulted in significant inhibition of tumor growth (De Palma M, Cancer Cell 2008). In the current work we report quite remarkable anti-leukemia effects, attaining durable complete responses in a sizable fraction of mice, in the absence of detectable signs of toxicity (in agreement with the previous characterization of the tolerability and selectivity of our delivery platform). These data, the mechanistic insights proving formally the reprogramming of the leukemia microenvironment at single-cell level, and the powerful synergy with other immunotherapy strategies presently receiving great attention, are in our view substantially novel findings that will advance previous work delivering IFN α . With full respect for the important work done by Belardelli over the last 20 years, our approach represents a new technology allowing durable delivery of IFN α into the tumor microenvironment, which can and will be translated into patients in the near future.

5. The model is also artificial because treatment was started day 5 after tumor cell injection. The extremely fast growth kinetics, referred to by the authors as “fast forward model”, probably leads very quickly to extremely high antigen amounts of foreign proteins and does not reflect the clinical situation the authors suggest to mimic.

We disagree with the reviewer's view on the “highly artificial” nature of our tumor model. Our B-ALL is a non-cell line model, does not grow in culture, is dependent on microenvironmental interactions and shows activated kinase signaling reminiscent of Philadelphia-like B-ALL in humans (Nucera et al, Cancer Cell 2016 and unpublished data). Murine CD19 CAR-T cell studies have been performed using CD19+ cell lines and the adoptive transfers of CAR-engineered T cells were performed at similar or even earlier times after tumor challenge as compared to our experimental setting (Kochenderfer et al, Blood 2010; Jacoby et al, Blood 2016; Ghosh et al, Nature Med 2017). Moreover, as pointed out above (see reply to the Reviewer's comment 3), although treatments were started “early”, control mice failed to benefit from any of the tested immunotherapies (see Fig.1c; Fig. 5a,d,j; Fig 6b,c and Supplementary Fig.1a, Fig. 9a,d,e; Fig. 13 a,b), thus speaking against the Reviewer's hypothesis that our model would facilitate immune responses because of extremely high antigen amounts of foreign proteins. In addition, in the new experiments employing CART19 cells we showed significant effect on tumor burden and improved long-term survival in IFN mice undergoing both early and late intervention trial (see **new Fig.6c and Fig.6g**). Finally, considering the high prevalence of the Philadelphia-like B-ALL subtype

in the adult population and the lower efficacy of CAR-T cell in adults as compared to the pediatric population, we regard our model as sufficiently mimicking the clinical situation, within the necessary constraints of a manageable experimental mouse model of the disease.

6. The gene signature experiments are descriptive. The role of M1/M2 macrophages, NK cells or NKG2D remains unclear.

We have now extended our original phenotypic analyses of immune cells in the tumor microenvironment (TME) by performing bulk and single cell RNA analysis on purified myeloid subpopulations from the leukemia-infiltrated tissues. Our new data provide substantial evidences that our IFN gene therapy reprograms the TME, imposing an immune-stimulatory gene signature and counteracting leukemia-induced expansion of immature immunosuppressive myeloid cells (**new Fig. 3, Supplementary Fig.5,6,7,8, Supplementary Online Excel Table 4,5,6**).

Summarizing from the new paragraph in the Results Section.

- Our RNA-seq analyses on purified tumor-associated macrophages revealed up-regulation of immunosuppressive genes (i.e. *Il10* and *Pdcd1* genes) and downregulation of genes involved in immune activation (i.e. downregulated genes were enriched in GO terms such as antigen presentation and leukocytes activation) in ALL-injected vs control mice.
- Conversely, IFN gene therapy in ALL-injected mice elicited an immune-stimulatory program characterized by up-regulation of IFN-Stimulated Genes, and genes enriched in GO terms related to defense response, leukocyte migration and response to interferon, and abrogated leukemia-induced up-regulation of *Il10* and down-regulation of MHC-II genes (**new Fig. 3a,b,c,d**).
- Importantly, whereas the transcriptomes of macrophages from tumor-free control and IFN mice showed high correlation, they were clearly distinct from those of ALL-bearing control mice (**new Supplementary Fig.5a,b**). Thus, RNA-sequencing analyses revealed leukemia-induced transcriptional changes in macrophages, which were substantially counteracted by IFN gene therapy.

In an attempt to further dissect the impact of the leukemia and IFN gene therapy on tumor infiltrating myeloid cells we performed single cell transcriptome analyses on splenic CD11b+ cells. This analysis revealed a major leukemia-dependent effect on the transcriptional landscape of non-classical monocytes (**new Fig.3e,f and Supplementary Fig.6a,b and Supplementary Online Excel Table 4,5**), which also underwent marked expansion in tumor-bearing mice (**see Fig.2b**). Tumor-associated non-classical monocytes up-regulated genes enriched in GO terms such as complement activation and negative regulation of inflammation, while they down-regulated genes linked to antigen processing and presentation (**new Fig.3g and Supplementary Online Excel Table 6**).

On the other hand, IFN gene therapy imposed an ISG-driven immune-stimulatory program to non-classical monocytes from ALL mice, as evidenced by up-regulation of genes enriched in GO terms related to defense and innate immune response, as well as MHC II genes (**new**

Fig.3g,h and Supplementary Online Excel Table 6). The changes described were relevant to the therapeutic benefits of gene therapy, as the extent of reprogramming observed at single-cell resolution correlated with leukemia inhibition.

Transcriptional reprogramming of the TME by IFN gene therapy was less effective in non-classical monocytes from mice that did not respond to IFN gene therapy (**new Supplementary Fig.8a**), as revealed by graph-based clustering and differential gene expression (**new Fig.3f,h and Supplementary Online Excel Table 5**).

Moreover, minimum-spanning tree (MST) analyses confirmed partial vs. effective reprogramming in cells from non responder vs. responder IFN mice (**new Fig.3i and Supplementary Fig.8b**). Overall, these data indicate that IFN gene therapy imposes an immune-stimulatory program to the myeloid cell infiltrate, conceivably priming towards activation of Th1 responses.

7. It is unclear where and how much IFN α is produced.

We and others have previously reported that the Tie2 gene is expressed on a sub-population of non-classical monocytes, hematopoietic stem cells (HSCs), endothelial cells and on a mesenchymal population of pericyte progenitors (Arai et al., 2004; De Palma M. et al., Nature Med. 2003; De Palma M. et al., Human Gene therapy 2003; De Palma M. et al., Cancer Cell 2005). Accordingly, upon transduction of hematopoietic progenitor stem cells (HSPCs) with the TIE2-GFP lentiviral vector we found expression of the transgene in monocytes (expression was higher in non-classical as compared to classical monocytes), tumor infiltrating monocytes/macrophages and in HSCs (De Palma M. et al., Cancer Cell 2008; Escobar G. et al., Science Translational Medicine). Despite the fact that the TIE2 promoter was also active in HSPC, when we used the TIE2-IFN vector to engineer mouse hematopoiesis we observed no evident toxicity, most probably because of the weak activity of the promoter. Accordingly, no IFN protein could be detected in the serum of the mice by ELISA (De Palma M et al., Cancer Cell 2008). However, it has been shown that chronic high level IFN exposure in the bone marrow has detrimental effects on hematopoiesis and may lead to hematopoietic stem cell exhaustion (Essers et al., Nature 2009; Sato et al., Nature Medicine 2009; Hartner et al., Nature Immunology 2009; King et al., Blood 2011). In line with these studies, we previously showed that mice transplanted with HSPC transduced with a vector expressing the IFN transgene ubiquitously in all hematopoietic cells (from a constitutive strong promoter), die shortly after transplant due to failure of hematopoietic reconstitution and overt bone marrow aplasia. Moreover, upon systemic expression of IFN from a transgene delivered to the liver we observed progressive weight loss, myelotoxicity with marked thrombocytopenia. Thus, in order to improve the safety of our delivery platform, we have exploited the miRNA-based post-transcriptional regulatory system to achieve a more stringent control of transgene expression. Specifically, we have endowed our vector platform with target sequences perfect complementary to miR-126 and-130a, both of which are expressed in HSPCs but not in the myeloid and lymphoid differentiated progeny (Gentner B. et al., Science Translational Medicine 2010). With the new vector, we showed that we could de-target transgene expression from HSPC while maintain its selective expression in differentiated myeloid cells (Escobar G., et al., Science Transl. Med. 2014).

IFN α is thus released within myeloid cell-infiltrated tissues, where an IFN response gene signature is detected, as shown in this manuscript, and such response is enhanced in the presence of leukemia, which expands the targeted population of non-conventional monocytes (**new Fig.3g,h and Supplementary Online Excel Table 6**). For obvious technical difficulties, and the conceivably low concentration, we could not measure the actual amount of IFN locally released within the leukemia-infiltrated tissues.

8. Countless numbers of publications have shown that mice, which rejected a cancer cell inoculum, are subsequently protected from a second challenge.

This is exactly why we used this well-established assay to prove that our IFN gene promoted the development of protective memory responses against OVA as well as other TAA, as shown by rejection of the parental OVA negative leukemia (Fig.4c,d,e and Supplementary Fig.16).

Minor Points:

The discussion is rather long.

We have now reduced the length of the discussion.

ATT is usually not performed with naïve but effector T cells. What happens in their IFN α model if effector OT-I are transferred?

The use of naïve T cells was rationally chosen to study the effect of IFN gene therapy also in the early steps of T cell activation/priming which could not be captured when using activated effector T cells.

In order to address the reviewer comment, we have now expanded our studies and demonstrated a major benefit of IFN gene therapy also on activated effector CART cells when applied to the parental (OVA-negative) leukemia (**new Fig.6, Supplementary Fig.14,15**). Similar to what found with naïve OT-I cells in the OVA-ALL model we have shown that IFN gene therapy enhances CART cell effector functions and persistence, CAR expression and overcomes the otherwise prompt acquisition of an exhaustion phenotype by the adoptively transferred T cells.

Discussion line 214, “neo-antigen that help to drive the transformed phenotype” is misleading since the driver is the mir-126 RNA and not one of the xenogeneic tumor antigens.

We have now removed this sentence.

Supplementary 4b lacks open symbols.

We have corrected the figure in the revised manuscript (now Supplementary Fig.12b).

Reviewer #2 (Remarks to the Author):

The authors utilize a gene transfer technique to illustrate enhanced anti-tumor efficacy in the context of IFN α and is further enhanced by CTLA-4 blockade. The authors, utilizing OVA modified ALL tumor cells, show enhanced anti-tumor efficacy in the context of IFN α (through genetically engineered macrophages and monocytes) mediated by both endogenous T cells (primarily reactive to OVA) as well as OVA specific OT-1 T cells. The authors further demonstrate IFN mediated changes within the tumor microenvironment with increased M1 macrophages and Th1 responses. Utilizing this ALL model, the authors further demonstrate epitope spreading in the context of CTLA-4 blockade, a phenomenon which apparently is predictive of tumor eradication. Collectively, the authors argue that a gene therapy approach with IFN α may have clinical application in part allowing for recruitment and activation of T cells specific to patient specific tumor neo-antigens.

Critiques

1. Targeted delivery of pro-inflammatory cytokines is not a novel concept (see IL-12 modified T cells in the context of pmel studies and melanoma targeted TILs). In particular, IL12 studies similarly demonstrated changes in the tumor microenvironment by locally delivered pro-inflammatory cytokines.

The framework of our studies lies in the translation of a myeloid-targeted, tumor-selective IFN α gene/cell therapy platform to patients suffering from hematologic malignancies. The demonstration that our B-ALL model induces prominent changes in the orthotopic immune microenvironment towards an immune paralytic state, and that targeted IFN gene therapy reverses these changes was not obvious and provides highly relevant information for moving towards a first-in-human clinical in the near future.

Importantly, we have now generated a substantial amount of new data providing novel insights on the activity of myeloid cell-based delivery of IFN into the leukemia microenvironment (**new Fig.3, Supplementary Fig.5,6,7,8, Supplementary Online Excel Table 4,5,6**, further discussed in the reply to the Reviewer comment 3). Moreover, we introduced a novel, clinically- relevant model of T cell-based immunotherapy reproducing the findings we previously obtained with the OT-1 model, i.e. a remarkable increase in efficacy

mediated by our IFN delivery strategy, using CAR19 T cells against the parental B- ALL model.

As shown in the **new Fig.6 and Supplementary Fig.14,15**, we found only a modest activity of this CART19 in control animals, at least in our experimental conditions, probably due to the aggressive nature of our non-cell line based B-ALL model, a relatively minor degree of T lympho-depletion following cyclophosphamide conditioning in the mouse (as opposed to humans), and ex vivo T cell engineering not fully optimized for mouse cells. Importantly, under the same conditions, IFN gene therapy-treated mice demonstrated substantial activity leading to the cure in up to two thirds of the mice. CART19 cells in responders showed transient (as opposed to persistent) Lag3 upregulation and signs of metabolic activation, providing new evidence that our IFN treatment renders the tumor microenvironment more permissive to anti-tumor T cell activity.

This effect was seen in mice chronically exposed to cell-based IFN delivery and persisted long-term after tumor challenge, strongly arguing against the induction of negative feedback to IFN mediating therapy resistance in our model, neither in the context of TAA-specific T cells, CART cells nor anti-CTLA4 treatment. The principle we describe here on 2 models is highly relevant, since it is generally applicable to T cells targeted to the tumor. Tumor targeting can be achieved in multiple ways, including pharmacologic treatment with bispecific T cell engagers (BITES). These strategies are increasingly entering the clinics, yet they still do not reach the efficiency of CART treatments.

Last but not least, we do not think that studies based on IL12 delivery can be considered overlapping with studies exploiting IFN α , even if one were using the same delivery strategy (which is not the case) nor the same experimental disease model (which also is not the case).

2. There is some concern about the highly artificial nature of this immune competent tumor model using a non-self OVA antigen which is highly immunogenic and fails to reflect spontaneous tumors in the clinical setting. Thus, despite the improved outcomes seen in these studies, the highly artificial nature of the tumor makes it questionable whether this data would translate well to the clinic.

We agree with the Reviewer concern that OVA antigen may not well recapitulate endogenous tumor neo-antigens which only carry mutations in few amino acids. We now explicitly mention this limitation in the discussion (lines 343- 347). However, in the original manuscript we also showed that IFN-gene therapy significantly inhibits the growth of parental OVA-negative leukemia (now Fig. 1c). We sustain that the B-ALL model used (Nucera et al, Cancer Cell 2016 and unpublished data from the Gentner lab) faithfully reproduces the disease encountered in adult patients, well ahead of other B-ALL models employed for immune-oncology studies and consistent with the increased difficulty we encountered to achieve responses, e.g. with the CD19 CART strategy, as compared to others. Importantly, whereas CTLA4-blockade therapy failed to show any significant effect in control mice injected with the parental ALL, it significantly inhibited leukemia growth and

improved mice survival when combined with IFN gene therapy (now Fig 1c and Supplementary Fig.1a).

Of note, leukemia is per se not sufficiently immunogenic to allow immune-mediated disease control, but can be made susceptible to immune pressure by transferring T cells targeted against leukemia antigens, allogeneic transplantation or, conceivably, by the new strategy reported in this manuscript. While leukemia is not a good model per se to study endogenous antigens (we would consider doing this for selected solid cancer models, which are not the subject of this work), our data suggest that IFN gene therapy has the potential to broaden anti-leukemia responses to less immunogenic tumor-associated antigens, which may be key to avoid immune escape. We sustain that OVA is a valid model system widely used in immuno-oncology studies that fulfills its purpose in our leukemia model.

3. Gene expression profile data is confusing and almost can be considered raw data. These plots should be removed and replaced by a more interpretable and more fully analyzed summary.

We have now replaced the original Nanostring data and original phenotypic analyses of immune cells in the tumor microenvironment (TME), with bulk and single cell RNA analysis on purified myeloid subpopulations from the leukemia-infiltrated tissues. Our new data provide substantial evidences that our IFN gene therapy reprograms the TME, imposing an immune-stimulatory gene signature and counteracting leukemia-induced expansion of immature immunosuppressive myeloid cells (**new Fig.3, Supplementary Fig.5,6,7,8, Supplementary Online Excel Table 4,5,6**). Summarizing from the new paragraph in the Results Section

- Our RNA-seq analyses on purified tumor-associated macrophages revealed up-regulation of immunosuppressive genes (i.e. Il10 and Pdcd1 genes) and downregulation of genes involved in immune activation (i.e. downregulated genes were enriched in GO terms such as antigen presentation and leukocytes activation) in ALL-injected vs control mice.
- Conversely, IFN gene therapy in ALL-injected mice elicited an immune-stimulatory program characterized by up-regulation of IFN-Stimulated Genes, and genes enriched in GO terms related to defense response, leukocyte migration and response to interferon, and abrogated leukemia-induced up-regulation of Il10 and down-regulation of MHC-II genes (**new Fig.3a,b,c,d**).
- Importantly, whereas the transcriptomes of macrophages from tumor-free control and IFN mice showed high correlation, they were clearly distinct from those of ALL-bearing control mice (**new Supplementary Fig.5a,b**). Thus, RNA-sequencing analyses revealed leukemia-induced transcriptional changes in macrophages, which were substantially counteracted by IFN gene therapy.

4. In tumor re-challenge studies (figure 4a-d), it would be of great interest if the authors could study the T cells in greater depth (i.e. more directly demonstrate antigen spreading) and even study whether adoptive transfer of T cells from these mice are able to eradicate both

OVA+ and OVA- tumors in tumor naïve mice. Further, in the text the authors point out that these mice were resistant to a challenge of a mix of OVA+ and OVA- tumor but this data is not provided.

The data requested by the Reviewer were already present, at least in part, in the original manuscript, and may have escaped her/his attention. In the original manuscript, we showed that IFN mice surviving a challenge with OVA-ALL cells were also protected against a challenge with parental OVA-negative leukemia or a mix of OVA+ and OVA- cells. These data are provided in Fig 4 c-e of the revised manuscript (Fig. 4b-d of the original manuscript). These figures show that long-term surviving IFN mice from the exp shown in Fig 4a were protected and survived from a re-challenge with both a mix of OVA+ and OVA- cells (Fig 4d of the revised manuscript) or, more impressively, with 100% parental OVA-negative cells (Fig.4e of the revised manuscript). Moreover, in two additional experiments also present in the original manuscript, we further showed that mice surviving long-term to the first challenge with OVA-ALL cells did show reactivity towards multiple tumor antigens beside Ovalbumin (see **Supplementary Fig.11a,b** of the revised manuscript) and that immune reactivity towards multiple antigens can predict mouse survival (see **Fig.5e** of the revised manuscript and **Supplementary Fig.16**).

In order to further strengthen the conclusion that IFN gene therapy promotes T cell priming against multiple tumor antigens, we have now performed a new experiment (OVA+ B-ALL in IFN gene therapy vs control mice, in the absence of CTLA4-blockade, see mice described in the **new Fig.4f**) to assess T cell reactivity against TAAs early after tumor injection, by measuring immune responses towards tTA, OFP or OVA epitopes by IFN-gamma ELISPOT assay and retrospectively correlated it to mouse survival (**new Fig.4f and Supplementary Fig.16**). We found that the two IFN mice that eventually survived the tumor challenge showed early reactivity against both OVA and tTA, confirming that immune response to multiple TAAs appears to be required for durable protection. Conversely, control mice showed immune reactivity against one or no antigens and none of them survived the tumor challenge. These new data further substantiate the results obtained in Fig. 4b-c of the original manuscript (**now Fig.4c-e of the revised manuscript**), which showed that IFN mice surviving the OVA-ALL challenge were also protected from a re-challenge with parental ALL cells. Overall, we now show that antigen spreading, experimentally tested either by re-challenging the mice with parental OVA-negative ALL (Fig. 4c-d) or by IFN-gamma ELISPOT assay (**Fig.4f and Supplementary Fig.16**) occurred in 8/41 (19.5%) IFN mice and only 1/43 (2.3%) controls (only mice in which immune reactivity was experimentally tested are taken into consideration for this analysis; $p=0.0136$, Fisher Exact test). All these mice showed long-term survival.

5. With respect to studies presented in figure 4h, the authors again note that these mice, when re-challenged a mix of OVA positive and negative tumors, a majority of mice survived. Again, this data is curiously not provided.

These data were provided in the **Supplementary Table 1** of the original manuscript (now **Supplementary Fig.16**).

6. Was statistical analysis conducted on the data presented in figure 4j-k? If not, why not?

In Fig. 4J (now **Supplementary Fig.11c**) we tested and showed the T cell immune reactivity by IFN-gamma ELISPOT assay against OVA, tTA, NGFR and OFP in control or IFN mice treated with anti-CTLA4 blockade therapy. Results from this analysis are used to stratify mice into the two groups (≥ 2 TAA and ≤ 1 TAA) as shown in the Kaplan-Meier curve from Fig.4k (now **Fig.5k** of the revised manuscript). In this latter analyses we showed that immune reactivity towards two or more antigens predicts mouse survival. We have now performed statistical analysis on these data by applying the Mantel-Cox test ($p=0.002$) to formally prove that mice showing immune reactivity against 2 or more antigens have a significantly improved rate of long term survival.

7. There was an excessive amount of relevant data presented in the supplementary data making review of the manuscript difficult.

We realize that our manuscript is data intense, and we had to make a choice in order to comply with journal space restrictions. We apologize if this resulted in a difficult review process. We have now re-organized the revised manuscript, attempting to reach a good compromise in distributing data between main and supplementary material.

8. It is not clear how the authors propose to translate this approach to the clinical setting. Do the authors propose that patients undergo an autologous bone marrow transplant with IFNa modified stem cells after initial chemotherapy? I am not sure this is a clinically feasible approach for a vast majority of patients. This needs to be better clarified.

A phase I/II clinical trial is planned to start soon in our institute. The data reported in this manuscript provides a biological rationale for this upcoming first-in-human trial of a cancer immunotherapy, and we have completed a full preclinical data package consisting of toxicity and biodistribution studies performed under “good laboratory practice” conditions supporting clinical translation. With regards to the clinical trial protocol design, we have held a pre-inquiry meeting with the Italian regulatory agency and already organized 2 advisory board meetings, gathering international key opinion leaders in the field. We have thus verified the clinical feasibility of our approach in different disease contexts and created a network of collaborators that will help enrolling the patients and conducting the trial. In synthesis, enrolled patients will undergo multi-modal tumor de-bulking, collect stem and progenitor cells for manufacturing of our drug product (DP) consisting of autologous CD34+ cells transduced with the human Tie2.IFNa.126T vector and receive DP infusion following a non-myeloablative conditioning chemotherapy based on alkylators (which, collaterally, have anti-

tumor activity). Given that a transduced cell chimerism as low as 5% in vivo has shown efficacy in specific dose-de-escalation studies (data to be submitted for publication elsewhere), our approach is clinically feasible for the majority of patients. We briefly mention how we conceive translating the result of this study into the context of a clinical application at the end of the Discussion.

Reviewer #3 (Remarks to the Author):

Escobar et al present data showing that genetically driven gene-based IFN α delivery in a B-cell ALL model inhibits tumor growth and modulates the tumor microenvironment resulting in a more effective antitumor immune response. They suggest that combining this approach with CTLA4 blockade or adoptive transfer of tumor antigen specific T-cells results in improved survival in a murine model.

Specific comments –

1. A primary point of the manuscript is that the interferon gene delivery is targeted and produced by tumor infiltrating monocytes and macrophages. Previous models using this approach have been in solid tumors (glioma, breast, colorectal cancers) and not hematological malignancies which have continuous exposure to the peripheral circulation. IFN α has been used as therapy in various hematological malignancies with clinical benefit when administered systemically. It would be important to show that similar results would not be seen if interferon alpha was simply systemically administered. Furthermore, the authors suggest that this approach could be developed clinically. To do that, it would be important to show that IFN α transfected monocytes can be infused resulting in similar effects.

As the reviewer is alluding to, a key rationale for our Tie2 promoter/enhancer-driven local IFN α delivery is an improved therapeutic window as compared to systemically administered recombinant IFN α protein. We have several lines of evidence that our gene-based strategy has a substantially reduced potential for systemic toxicity, a major limitation of recombinant IFN α administration to patients.

(1) Systemic in vivo injection of lentiviral vector expressing IFN α from the housekeeping PGK promoter was performed in a previous study to transduce splenic and liver cells, which became a source of transgene production. Using this approach, we achieved high level of IFN α in the sera of the mice (418 ± 124 pg/ml) which was associated with myelotoxicity, weight loss and thrombocytopenia, but -opposite to the Tie2.IFN α gene therapy approach which did not result in detectable IFN α levels in the serum- lack of therapeutic effect in a glioma tumor model (De Palma M, Cancer Cell 2008).

(2) Transplantation of HSPC transduced with a PGK.IFN vector ubiquitously expressing moderate levels of IFN α in all hematopoietic cells leads to engraftment failure and death of

the mice, highlighting the importance of controlling both IFN α expression levels and specificity of expression restricted to a narrow and pathogenetically relevant cell population, such as TEMs in the case of the Tie2.IFN α .126T construct.

(3) Systemic injection of a half-life optimized recombinant murine IFN molecule showed similar short-term efficacy in restricting growth of a standard subcutaneous grafts of cancer cell lines as our gene therapy approach at an in vivo chimerism of as low as 20%. However, the systemic treatment caused substantially more myelosuppression than the gene therapy approach, not to speak of systemic tolerability issues which are notoriously hard to assess in the mouse model. Moreover, mice treated with systemic IFN α became progressively insensitive to the treatment, while IFN α gene therapy treated mice showed evidence for stable, long term IFN α activity. These studies are currently being completed and will be assembled into a separate manuscript.

Regarding clinical development, we aim to develop an HSPC-based delivery approach similar to the one described in this manuscript (see response to reviewer #2, point 8, for a more detailed elaboration of the clinical strategy). Concerning the possibility to engineer monocytes to selectively delivery IFN into the tumors, we have doubts on the homing and durability of these differentiated cells upon systemic delivery.

2. The manuscript highlights the importance of effector T-cells and suggests that the IFN α delivery modulates the microenvironment to promote T-cell function. The OVA-ALL model used does not seem to be very immunogenic and it would appear that OFP and tTA that are also expressed as neo-antigens after transfection of the cells may also be responsible for CD8 $^{+}$ cell activation. The authors should clarify which of these antigens is the most important. Furthermore, when CD8 $^{+}$ cells are depleted from the model, there is only a modest decrease in tumor burden suggesting that other mechanisms could be important. They suggest a slower proliferation rate of the tumor cells due to IFN α could be the cause. This should be confirmed. They further state that the delay in tumor cell proliferation may allow for the expansion of tumor-specific CTL but provide no data to support this. This should be done.

Among the mice that survived long term to the tumor challenge and were tested for immune reactivity against both the OFP and tTA antigens by the IFN-gamma ELISPOT assay we found that nearly 78% (n=14/18) of them showed reactivity against the tTA antigen whereas only 33% (n=6/18) of them showed reactivity against the OFP protein (see **Supplementary Fig.16**). Thus, these results suggest that the tTA protein may be more immunogenic and/or more relevant to induce long-term durable anti-tumor responses in the mice. Still, OVA is triggering the strongest immune response, as suggested by the higher frequency of IFN γ spots in the IFN group as compared to OFP or tTA (see e.g. Fig. 4f of the revised manuscript). The fact that the reviewer notes that the OVA-ALL model does not seem to be very immunogenic highlights the potential of the disease to evade immunity (as suggested by our new bulk and single cell RNAseq analysis included in Figure 3 of the revised manuscript) thereby more faithfully reproducing the clinical behavior of leukemia in patients as compared to other published models.

Concerning the point whether IFN may affect ALL growth through additional mechanisms beside inducing adaptive immunity, we had shown this data in the Supplementary Fig.3a-f of the original manuscript. We have now clarified in the results that these two mechanisms contribute to the inhibition of ALL growth, although induction of adaptive immunity is by far the dominant one, as also mentioned in the Reviewer's words: "*when CD8+ cells are depleted from the model, there is **only a modest decrease** in tumor burden suggesting that other mechanisms could be important* (whereas in our view the fact that there is only a modest decrease in tumor burden when CD8+ cells are depleted from the model, as compared to the substantial effect seen in mice which retain CD8+ cells, indeed support the contention that CD8+ cells are responsible for most of the effect.)

3. The authors report that OVA-specific T-cells upregulate LAG-3 and acquire a memory phenotype. When studied at the time of sacrifice, cells in IFN mice had downregulated PD-1 and LAG-3 but maintained the memory phenotype. They suggest that IFN α exposure prevented T-cell exhaustion. Given that PD-1 expression is also typically seen when cells are activated, it would be important to confirm that IFN α is protective against T-cell exhaustion by treating control OVA-specific T-cells ex-vivo and showing the same thing.

It is known from the literature that, whereas transient upregulation of PD1 and LAG3 normally occur during T cell activation, stable and persistent expression of these molecules has been associated with T cell dysfunction in both chronic viral infection and tumor models (Schildberg F.A. et al., Immunity 2016). In our study, we have shown that IFN gene therapy can prevent persistent PD1 and Lag3 expression on the surface of OT-I T cells and improve their tumor killing capacity in vivo. In the revised version of the manuscript we have further validated these results by employing activated CART19 cells. Mechanistic analyses indicate that IFN gene therapy enhances T cell activation, CAR expression and overcomes the otherwise prompt acquisition of an exhaustion phenotype by the adoptively transferred T cells. These data are shown in the **new Fig.6, Supplementary Fig.14,15**. Overall, we believe that although a direct effect of IFN may contribute to prevent T cells exhaustion, our IFN gene therapy strategy exerts a broader effect on different cells in the tumor microenvironment to reprogram them in a state more permissive to the deployment of effector T cell responses. In this regard we have now performed bulk and single cell RNA analysis on purified myeloid subpopulations from the leukemia-infiltrated tissues and provided stringent evidence that our IFN gene therapy reprograms the TME, imposing an immune-stimulatory gene signature and counteracting the expansion of immature immunosuppressive myeloid cells driven by the leukemia (**new Fig.3, Supplementary Fig.5,6,7,8, Supplementary Online Excel Table 4,5,6**) thus favoring the induction of more effective T cells responses.

4. To enhance the IFN effect, the authors combined tumor targeted IFN α treatment with CTLA4 blockade. They should clarify why an anti-CTLA4 approach was chosen when their

data suggests a role for PD-1 or LAG-3 blockade as upregulation of LAG-3 and PD-1 were seen in hypofunctional effector cells.

The flow of text in the revised manuscript now better clarifies the rationale for testing CTLA4 blockade.

Editorial Note: this version of the manuscript has also been previously reviewed at another journal that is not operating a transparent peer review scheme. This document only contains reviewer comments and rebuttal letters for versions considered at Nature Communications.

REVIEWERS' COMMENTS:

Reviewer #1 (Remarks to the Author):

The authors provide a lengthy reply but the main critic has not been resolved. I only repeat and specify the most critical points.

1. The study lacks novelty.

Given the many publications by the authors and on IFN α in general, there is little novelty.

2. The model is highly artificial.

It remains that only surrogate antigens are analyzed; long open reading frames with multiple MHC I and MHC II epitopes. Does not occur in leukemia. The authors still hide this fact and call them tumor-associated antigens (TAA). This is wrong, TAA are self-antigens, ova or tTA are tumor-specific but are surrogate antigens.

3. A countless number of publications investigated ovalbumin as surrogate tumor antigens with similar or even better therapeutic effects.

The authors still ignore that large established tumors targeting ova were rejected in other studies (Engels et al, Cancer Cell 2013). In terms of efficacy, the current model does not compare to earlier studies.

5. The model is also artificial because treatment was started day 5 after tumor cell injection. To say it more clearly: in a clinical setting, the immune system is exposed to tumor antigens for a very long time resulting in profound tolerance. In a five-day tumor model, the therapeutic modality jumps in into the T cell priming phase. Additionally, there is the artificial process of tumor cell injection, facilitating T cell activation. The authors' model could not be more artificial.

7. It is unclear where and how much IFN α is produced.

Authors provide a lengthy reply but question still open: where and how much?

8. Countless numbers of publications have shown that mice, which rejected a cancer cell inoculum, are subsequently protected from a second challenge.

The authors did not get the point: In early days, tumor immunity was induced by life tumor challenge and removal of the tumor by ligation. IFN α prevents tumor take but it remains unclear whether it directly contributes to T cell immunity.

The novel CD19-CAR experiment does not relate to the rest of the study. Effects of IFN α are moderate. Importantly, CD19-CAR therapy is so effective in the clinic that it certainly will never be combined with the complicated setting used by the authors.

Reviewer #2 (Remarks to the Author):

The authors have adequately addressed my concerns

Reviewer #3 (Remarks to the Author):

The authors have adequately addressed my previous critiques

Reviewer #4 (Remarks to the Author):

I have previously reviewed this manuscript for a prior submission. The revised manuscript submitted here, addresses my prior concerns. I have reviewed the comments from the other reviewers and the response of the authors. My assessment is that while there is more to be done (as there always is), the authors have demonstrated sufficient novelty and significance for the journal to which the manuscript is now submitted. While many of the parts to the manuscript have been described previously, it is often important to then determine how the parts can be put together and what the consequences are. My assessment is that the authors have capitalized on their expertise in lentiviral delivery and lineage specific expression of transgenes and in this work demonstrated both the promise and limits of the approach of using gene therapy to express Ifn in the tumor microenvironment. Certainly the work would be strengthened by the authors studying antigen spread to true neo-antigens rather than to artificial neo-antigens but I support the assessment that would go beyond the scope of this work.

Response to Reviewers' comments:

The authors provide a lengthy reply but the main critic has not been resolved. I only repeat and specify the most critical points.

1. The study lacks novelty.

Given the many publications by the authors and on IFN α in general, there is little novelty.

We respectfully disagree with the Reviewer's opinion on this point, as already argued in our previous response.

2. The model is highly artificial.

It remains that only surrogate antigens are analyzed; long open reading frames with multiple MHC I and MHC II epitopes. Does not occur in leukemia. The authors still hide this fact and call them tumor-associated antigens (TAA). This is wrong, TAA are self-antigens, ova or tTA are tumor-specific but are surrogate antigens.

We now use the term surrogate Tumor-Specific Antigen (TSA), as suggested by the Reviewer, instead of TAA throughout the manuscript and have further toned down some claims on the expected translational value of the work emphasizing the necessarily experimental nature of the model used to investigate induction and deployment of tumor-specific adaptive immunity (see highlighted new text from abstract and discussion below). We had already mentioned in the prior version of the manuscript the limitations of experimental TSA commonly used in experimental cancer models to mimic neo-antigen arising in spontaneous tumor, as can be seen in the paragraph from the Discussion reported below.

Abstract: This reprogramming promotes T-cell priming and effector function against multiple surrogate tumor-specific antigens, inhibiting leukemia growth in our experimental model.

*Discussion: The rapid *in vivo* induction of robust immunity against multiple surrogate TSAs by our strategy may have translational implications for cancer immunotherapy, albeit with the caveat that TSAs commonly used in experimental cancer models are xenogeneic proteins and provide only a surrogate of clinically relevant neo-antigens, which are typically altered self-proteins carrying few amino acid substitutions and are thus likely to be less immunogenic. ... We should also mention that, whereas our IFN gene therapy also inhibited the growth of the parental ALL, the introduction of a dominant TSA such as OVA might favor an initial cytotoxic response robust enough to allow effective spreading of the immune repertoire to multiple surrogate TSAs and*

establish durable protection. It is possible that the requirement for a strong TSA might reflect the very rapid course of the disease in the transplant setting and may not apply to spontaneous tumors arising in patients, **where on the other hand the immune system is exposed to TSAs for a long time resulting in profound tolerance.**

3. A countless number of publications investigated ovalbumin as surrogate tumor antigens with similar or even better therapeutic effects. The authors still ignore that large established tumors targeting ova were rejected in other studies (Engels et al, Cancer Cell 2013). In terms of efficacy, the current model does not compare to earlier studies.

We still do not understand this comment. The study cited by the Reviewer used the well-established fibrosarcoma cell line MC57 transduced to overexpress various TSAs, providing an elegant investigation of adaptive immunity to surrogate TSA and highlighting the importance of targeting peptides with high affinity for MHC class I when designing T cell-based immunotherapy. One cannot compare the actual “efficacy” of antitumor responses among different models, in particular considering that we used a transplantable early passage leukemia arising in mice and which only grows in vivo.

5. The model is also artificial because treatment was started day 5 after tumor cell injection. To say it more clearly: in a clinical setting, the immune system is exposed to tumor antigens for a very long time resulting in profound tolerance. In a five-day tumor model, the therapeutic modality jumps in into the T cell priming phase. Additionally, there is the artificial process of tumor cell injection, facilitating T cell activation. The authors’ model could not be more artificial.

As mentioned in point 2 above, we have further toned down some claims on the expected translational value of the work emphasizing the necessarily experimental nature of the model used to investigate induction and deployment of tumor-specific adaptive immunity (see highlighted new text from discussion below). Short of using spontaneously arising tumor models in transgenic mice, which have also limitations on their own in terms of faithfully representing naturally arising tumors in humans, injection of syngeneic or allo/xeno-geneic tumor cells is unavoidable to perform manageable experiments of tumor treatment, as shown by the vast preponderance of this approach in the relevant scientific literature.

From the Discussion: It is possible that the requirement for a strong TSA might reflect the very rapid course of the disease in the transplant setting and may not apply to spontaneous tumors arising in patients, **where on the other hand the immune system is exposed to TSAs for a long time resulting in profound tolerance.**

We already mentioned in a previous response the relevant features of our B-ALL model in mimicking the human disease, as published in a recent paper of ours cited in the Introduction (Nucera, S. et al. Cancer Cell 29, 905–921, 2016).

7. It is unclear where and how much IFN α is produced.

Authors provide a lengthy reply but question still open: where and how much?

As mentioned in our previous response, we could not measure by antigen capture the actual amount of IFN locally released within the leukemia-infiltrated tissues. Our previously published studies on the development, validation and testing of the IFN α gene therapy platform provide data showing the specific induction of IFN responsive genes at tumor sites with much lower to undetectable response in other tissues. Detailed studies have been performed in normal mice (without tumors) to characterize the dose dependence and pharmacokinetic of our gene therapy product in preparation for clinical testing and will be reported elsewhere.

8. Countless numbers of publications have shown that mice, which rejected a cancer cell inoculum, are subsequently protected from a second challenge. The authors did not get the point: In early days, tumor immunity was induced by life tumor challenge and removal of the tumor by ligation. IFN α prevents tumor take but it remains unclear whether it directly contributes to T cell immunity.

We still do not understand this comment. Our data clearly show that IFN induces increased numbers and more active TSA-specific T cells and that IFN-dependent tumor protection is mediated by effector CD8 T cells (see for instance Fig. 2d, where anti-tumor effects are abrogated by depleting CD8 T cells). As mentioned in the Discussion, IFN effects are likely pleiotropic and exerted both on myeloid and lymphoid cells, thus necessarily implying both direct and indirect mechanisms of action contributing to the induction and deployment of tumor specific T cell immunity.

The novel CD19-CAR experiment does not relate to the rest of the study. Effects of IFN α are moderate. Importantly, CD19-CAR therapy is so effective in the clinic that it certainly will never be combined with the complicated setting used by the authors.

Even the most enthusiastic supporter of CAR-T cell therapy would agree, as we argue in the Introduction that: “an immunosuppressive TME represents a major impediment towards successful immunotherapy, especially against solid tumor masses”, and that there is urgent need of novel strategies aimed at - taken from our Discussion: “enhancing the efficacy of adoptive T cell immunotherapy and broaden its reach to tumors with low mutational load or lacking dominant neo-antigens, and to solid tumors where T cell penetration and effector activity is often rate-limiting”.

Furthermore, we disagree that effects of IFN α are moderate, as long-term survival is extended from nearly none with CAR-T cells alone up to 70% when CAR-T cells were combined with IFN gene therapy.